# Bistability of a coupled Aurora B kinase-phosphatase system in cell division

**Anatoly V Zaytsev[1†], Dario Segura-Peña[2†], Maxim Godzi[1,3], Abram Calderon[2], Edward R Ballister[2], Rumen Stamatov[1], Alyssa M Mayo[2], Laura Peterson[4,5], Ben E Black[6], Fazly I Ataullakhanov[3,7,8*], Michael A Lampson[2*], Ekaterina L Grishchuk[1*]**

[1]Department of Physiology, Perelman School of Medicine, University of Pennsylvania, Philadelphia, United States; [2]Department of Biology, University of Pennsylvania, Philadelphia, United States; [3]Center for Theoretical Problems of Physicochemical Pharmacology, Russian Academy of Sciences, Moscow, Russia; [4]Department of Biology, Massachusetts Institute of Technology, Cambridge, United States; [5]Department of Chemistry, Massachusetts Institute of Technology, Cambridge, United States; [6]Department of Biochemistry and Biophysics, Perelman School of Medicine, University of Pennsylvania, Philadelphia, United States; [7]Federal Research and Clinical Centre of Pediatric Hematology, Oncology and Immunology, Moscow, Russia; [8]Department of Physics, Moscow State University, Moscow, Russia

**\*For correspondence:**
ataullakhanov.fazly@gmail.com
(FIA); lampson@sas.upenn.edu
(MAL); gekate@mail.med.upenn.
edu (ELG)

[†]These authors contributed
equally to this work

**Competing interests:** The
authors declare that no
competing interests exist.

**Reviewing editor:** Jon Pines,
The Gurdon Institute, United
Kingdom

**Abstract** Aurora B kinase, a key regulator of cell division, localizes to specific cellular locations, but the regulatory mechanisms responsible for phosphorylation of substrates located remotely from kinase enrichment sites are unclear. Here, we provide evidence that this activity at a distance depends on both sites of high kinase concentration and the bistability of a coupled kinase-phosphatase system. We reconstitute this bistable behavior and hysteresis using purified components to reveal co-existence of distinct high and low Aurora B activity states, sustained by a two-component kinase autoactivation mechanism. Furthermore, we demonstrate these non-linear regimes in live cells using a FRET-based phosphorylation sensor, and provide a mechanistic theoretical model for spatial regulation of Aurora B phosphorylation. We propose that bistability of an Aurora B-phosphatase system underlies formation of spatial phosphorylation patterns, which are generated and spread from sites of kinase autoactivation, thereby regulating cell division.

## Introduction

Aurora B, a component of the chromosomal passenger complex (CPC), is an essential kinase that is highly enriched at different intracellular locations from which it regulates cell division: it localizes initially at the inner centromere and subsequently at the anaphase spindle midzone (*Carmena et al., 2012*). Accumulating evidence indicates that Aurora B is capable of phosphorylating substrates that are located at a significant distance from its major binding sites. In anaphase, a long-range phosphorylation gradient is established around the spindle midzone (*Fuller et al., 2008*; *Tan and Kapoor, 2011*), but extending well beyond major sites of kinase localization (*Figure 1A*). This phosphorylation gradient controls the stability and length of the central spindle (*Ferreira et al., 2013*; *Uehara et al., 2013*), chromosome decondensation and nuclear envelope reassembly (*Afonso et al., 2014*). Similar distance-dependent phosphorylation is observed prior to anaphase onset, but at this stage Aurora B localizes to chromatin with highest concentration at the inner centromere, where CPC binding sites are enriched. During metaphase the primary targets for Aurora B,

**eLife digest** Cell division is a highly organized process that involves a series of major changes. First, the cell's chromosomes are copied and arranged at the middle of the cell. Then, the pairs of copied chromosomes are separated and pulled towards opposite ends of the cell and, finally, the cell splits in two. These steps are mainly regulated by modifications to proteins, and enzymes called protein kinases play an important role because they add phosphate groups to, or phosphorylate, so-called 'substrate' proteins to change their activities. Other enzymes called phosphatases are also important because they remove the phosphate groups from the substrates to reverse the effects.

The kinase Aurora B is required for several steps during cell division and has been widely studied. This kinase is enriched in specific locations within the cell, for example at the centromere regions of the chromosomes as they line up at the cell's center. However, Aurora B phosphorylates substrates located at distant sites on the chromosome, with less phosphorylation at sites farther from the centromere. The level of phosphorylation also changes as chromosomes become aligned. Aurora B can activate itself and this ability was suspected to help this spatiotemporal regulation. However, it was not clear how the observed gradients of kinase activity might form.

Zaytsev, Segura-Peña et al. set out to answer this question by first mixing in a test tube purified Aurora B and an inhibitory phosphatase. This revealed that this kinase-phosphatase system is 'bistable', meaning that it has two stable states, low or high kinase activity, and that these states could switch in response to small changes in enzyme concentrations. Further experiments showed that this system has a kind of memory such that the level of activity (low or high) persists for a range of concentrations and depends on the system's prior history.

Zaytsev, Segura-Peña et al. then showed that both of these properties, the two stable states and the memory, exist in dividing human cells, and then went on to develop a mathematical model of how such bistability could set up gradients of Aurora B kinase activity. At the sites of highest concentration at the centromere, Aurora B can overcome inhibition by phosphatase and activates itself, as in the test tube. This activity spreads to more distant locations as active kinase molecules activate neighboring kinase molecules, establishing the areas with a high state of activity. As the local Aurora B concentration decreases further from the centromere, the phosphatase switches Aurora B into the low activity state, establishing a steep gradient of kinase activity in a region where its substrates that are important for chromosome segregation are located. Importantly, the shape and location of this gradient are predicted to depend on forces that stretch the lined chromosomes apart, offering a plausible mechanism to explain phosphorylation changes in response to tension. These theoretical insights and experimental approaches could be used to study other coupled kinase-phosphatase systems.

such as the microtubule-binding protein Hec1/Ndc80, are located hundreds of nanometers away at the outer kinetochore. As in anaphase, phosphorylation is lower on substrates positioned farther from Aurora B binding sites, indicating existence of a gradient of Aurora B activity (*Keating et al., 2009*; *Liu et al., 2009*; *Welburn et al., 2010*; *DeLuca et al., 2011*; *Suzuki et al., 2014*). Interestingly, changes of position of as little as 30–50 nm are associated with different levels of phosphorylation of both endogenous and exogenous Aurora B substrates at kinetochores (*Welburn et al., 2010*; *Suzuki et al., 2014*), indicating that the spatial regulation of Aurora B activity is very precise.

One model to explain such well-controlled long-range spatial activity is by a specialized pool of Aurora B localized in close proximity to its targets (*Krenn and Musacchio, 2015*). At the outer kinetochore, for example, the observed gradient of substrate phosphorylation could correspond to the outermost region of the localization gradient of chromatin-bound kinase (*Liu et al., 2009*), or reflect the ability of Aurora B to reach these substrates by an elongated INCENP tether, the CPC component that directly binds Aurora B and is important for its mitotic functions (*Samejima et al., 2015*). In this view, the less abundant but proximally located Aurora B pool plays a more physiologically important role than the distant centromeric pool. Support for the kinetochore pool model comes from experiments in budding yeast, which show that the centromere localized pool of Aurora B (Ipl1) can be removed without major consequences for mitotic progression (*Campbell and Desai,*

2013). In several other systems, however, disrupting CPC targeting to centromeres leads to strong mitotic defects (*Vader et al., 2006*; *Tsukahara et al., 2010*; *Wang et al., 2010*; *2012*; *Yamagishi et al., 2010*), suggesting that the centromere-localized pool is essential for normal cell division.

An alternative model to explain how Aurora B activity is controlled at distances away from its most abundant localization sites is that this pattern depends on a biochemical crosstalk between the bound Aurora B and its cytosolic pool, which recent quantitative measurements estimate as ~25% of total Aurora B (*Mahen et al., 2014*). Cytosolic gradients of another mitotic regulator, RanGTP, play important roles in regulating spindle assembly (*O'Connell and Khodjakov, 2007*; *Kalab and Heald, 2008*), and a similar mechanism could contribute to long-range Aurora B activity. In this reaction-diffusion model, activation of Aurora B takes place at sites with high kinase concentration, such as the inner centromere or anaphase spindle midzone (*Lampson and Cheeseman, 2011*). These sites exchange quickly with a cytosolic pool (*Fernández-Miranda et al., 2010*), so they could serve as a source of active kinase, which has been proposed to spread to distant targets via diffusion (*Fuller et al., 2008*; *Wang et al., 2011*). However, it is not clear whether a gradient based only on the diffusion of soluble activated kinase from the inner centromere could account for changes in Aurora B substrate phosphorylation within the length scale of the kinetochore (*Krenn and Musacchio, 2015*). In contrast, bistable reaction-diffusion systems can in principle exhibit complex spatial patterns and support sharp boundaries of system components (*Kapral and Showalter, 1995*; *Lobanova and Ataullakhanov, 2003*; *Liehr, 2013*). Bistable homogeneous systems (i.e., with mixing) can switch between the alternative states, characterized by high and low activity, with no intermediate states. Furthermore, unlike in regular trigger systems, in bistable systems the high and low states can co-exist, leading to hysteresis, when the output of the system depends on its prior history (*Martinov et al., 2000*; *Angeli et al., 2004*; *Tsyganov et al., 2012*; *Noori, 2013*).

Published results indicate that Aurora B kinase could in principle engage in complex non-linear behaviors. Most importantly, Aurora B can activate itself via phosphorylation of its activation loop and of a conserved TSS motif in the C-terminus of INCENP (*Bishop and Schumacher, 2002*; *Honda et al., 2003*; *Yasui et al., 2004*; *Sessa et al., 2005*; *Kelly et al., 2007*; *Xu et al., 2010*). Conversely, phosphatase can inactivate the kinase by dephosphorylating sites on Aurora B and INCENP (*Sessa et al., 2005*; *Kelly et al., 2007*; *Rosasco-Nitcher et al., 2008*), which could potentially help to shape the spatial gradient of Aurora B activity. Whether these reactions can lead to bistability in a coupled Aurora B-phosphatase system has not been investigated. Here, we examine the mechanisms that control Aurora B activity using cellular and simplified in vitro systems and mathematical modeling. First, we designed a novel molecular system to control Aurora B localization in cells, to directly test the importance of the centromeric pool of Aurora B in long-range activity. Second, we used purified components to reconstitute a simplified coupled Aurora B kinase-phosphatase system in vitro and showed that it exhibits bistability and hysteresis in the physiological range of Aurora B concentration. Because the complex, non-linear dynamics of reaction-diffusion systems and their spatial behavior are not intuitive, we constructed quantitative models to assist analysis of homogeneous biochemical reactions and formation of phosphorylation patterns in cells. We then developed experimental methods to analyze bistability and hysteresis of Aurora B-dependent phosphorylation in live mitotic cells, linking our biochemical findings with Aurora B regulation in cells. With these multiple approaches we provide strong evidence for a model in which spatiotemporal regulation of Aurora B is governed by a bistable reaction-diffusion mechanism.

## Results

### Concentrating Aurora B at centromeres leads to phosphorylation of distant chromatin substrates

Because experiments in budding yeast have raised questions about whether concentrating Aurora B at centromeres is necessary for its mitotic function (*Campbell and Desai, 2013*), we designed an experiment to measure phosphorylation in live human cells while manipulating Aurora B localization with temporal control. We made a cell line that inducibly knocks down endogenous INCENP, while expressing an INbox construct that can bind and activate Aurora B (*Sessa et al., 2005*) but does not interact with other CPC components. The Aurora B–INbox complex is sufficient for enzymatic activity

but does not localize to any particular intracellular structure because it does not form the full CPC. To control localization, we used rapamycin-based dimerization (*Putyrski and Schultz, 2012*), with FRB fused to INbox and FKBP fused to the centromere protein CENP-B (*Figure 1—figure supplement 1A*). FKBP and FRB are domains that dimerize in the presence of rapamycin. This system allows us to measure immediate effects in live cells within minutes of concentrating Aurora B at centromeres.

To monitor changes in Aurora B kinase activity at a distance from sites of localization at centromeres, we used a FRET-based biosensor targeted to chromatin by fusion to histone H2B (*Fuller et al., 2008*). When endogenous INCENP is replaced with INbox, which is freely diffusing in the cytosol, phosphorylation is uniformly low, indicating that the cytosolic kinase pool on its own is incapable of maintaining high kinase activity along chromosome arms. Addition of rapamycin led to INbox recruitment to centromeres within minutes, accompanied by sensor phosphorylation; importantly the signal was visible all over the chromatin (*Figure 1B*). For these experiments cells were arrested in mitosis with a kinesin-5 inhibitor, so that chromosomes were positioned radially around a monopolar spindle with centromeres oriented toward the center (*Mayer et al., 1999*). With this arrangement of chromosomes, a transient phosphorylation gradient was evident extending from centromeres, similar to previous experiments in which Aurora B activity was manipulated by global inhibition (*Wang et al., 2011*). Similar results were observed for cells arrested with nocodazole (*Figure 1—figure supplement 1B*). Thus, concentrating Aurora B at centromeres of a mammalian cell is necessary and sufficient to regulate kinase activity at distal cellular locations, warranting further investigation of the kinetic mechanisms of Aurora B autoactivation.

## Reconstitution of Aurora B kinase autoactivation in vitro demonstrates both cis and trans components

Highly concentrated centromeric kinase may become a source of active kinase for establishing spatial patterns if Aurora B can robustly activate itself in trans, i.e. intermolecularly (*Sessa et al., 2005*; *Kelly et al., 2007*; *Lampson and Cheeseman, 2011*). To determine the kinetic constants for Aurora B autoactivation, we measured phosphorylation in vitro in real time using purified recombinant Aurora B kinase with an INbox fragment, which is sufficient for kinase autoactivation (*Sessa et al., 2005*; *Rosasco-Nitcher et al., 2008*) (see Materials and methods and *Figure 2—figure supplement 1*). With purified kinase, the INCENP TSS motif, an established autophosphorylation site associated with kinase activation (*Bishop and Schumacher, 2002*; *Honda et al., 2003*; *Sessa et al., 2005*), was phosphorylated, as determined by immunoblotting with a phospho-specific antibody (*Salimian et al., 2011*; *Figure 2—figure supplement 1D*). This phosphorylated kinase was highly active, as shown using a chemosensor composed of a peptide containing an Aurora kinase substrate consensus site conjugated to a sulfonamido-oxine (Sox) fluorescent probe (*Figure 2—figure supplement 2*) (*Gonzáles-Vera et al., 2009*). Phosphorylation-induced increase in fluorescence of the chemosensor was followed in real time with a spectrofluorimeter, and the Michaelis–Menten, Lineweaver–Burk and Hanes–Woolf plots were analyzed (see Materials and methods), giving $K_M = 320$ μM and $k_{cat} = 19$ s$^{-1}$, similar to a previous report for Aurora A kinase (*Gonzáles-Vera et al., 2009*). To examine activity of Aurora B in the dephosphorylated state, we incubated the kinase with λ phage phosphatase, which has previously been reported to dephosphorylate INCENP (*Rosasco-Nitcher et al., 2008*), and observed loss of INCENP phosphorylation (*Figure 2—figure supplement 1D*). Phosphonoacetic acid was then added to inhibit the phosphatase (*Reiter et al., 2002*) and chemosensor phosphorylation was measured. The dephosphorylated Aurora B kinase was two orders of magnitude less active than the phosphorylated Aurora B, consistent with previous studies (*Eyers et al., 2005*; *Sessa et al., 2005*), so we refer to this kinase state as partially active.

Next, we sought to determine the kinetic parameters of Aurora B autoactivation. At 10–30 nM of partially active kinase, chemosensor phosphorylation was barely detected. This finding is consistent with our results using INbox replacement in cells with no rapamycin, since this low concentration range was reported for cytosolic Aurora B (*Mahen et al., 2014*). At 0.16–1.5 μM kinase, chemosensor phosphorylation increased nonlinearly with time, indicating autoactivation (*Figure 2A, Figure 2—figure supplement 2G*). Previous studies have reported that this autoactivation takes place in trans (*Sessa et al., 2005*; *Rosasco-Nitcher et al., 2008*) (*Figure 2B*), predicting that the coefficient for this increase vs. kinase concentration is close to 2 when plotted on a logarithmic scale. The measured slope in our experiments with low kinase concentrations was 1.23 ± 0.02

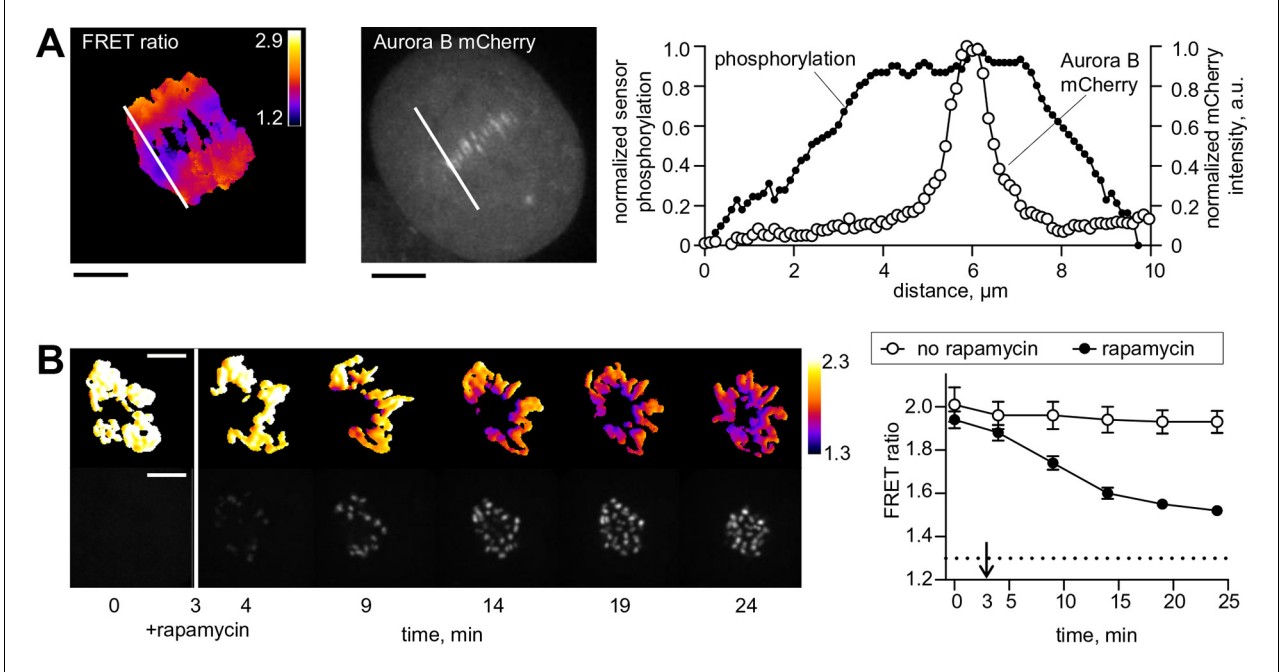

**Figure 1.** Spatial phosphorylation patterns in mitotic HeLa cells. (**A**) A HeLa cell expressing the chromatin-targeted Aurora B sensor and Aurora B-mCherry was imaged in anaphase, 10 min after addition of an Mps1 inhibitor, reversin, to increase occurrence of lagging chromosomes. The FRET ratio image shows the YFP/CFP emission ratio, color-coded as indicated. Scale bar is 5 µm. The plot shows normalized sensor phosphorylation (left axis) calculated from the FRET ratio data (see Materials and methods) and Aurora B localization signal (right axis) along the white lines which were drawn along the spindle axis in images on the left. (**B**) HeLa cells expressing CENP-B-FKBP, mCherry-INbox-FRB and miRNAs to deplete endogenous FKBP and INCENP, and the chromatin-targeted Aurora B sensor. Cells were treated with the kinesin-5 inhibitor STLC to generate monopolar spindles, then imaged live during rapamycin addition to induce INbox and Aurora B recruitment to centromeres. Images show INbox recruitment (bottom panels) and the YFP/CFP emission ratio (top panels) for one cell. Graph shows the FRET emission ratio averaged over chromatin in multiple cells (n≥10) treated at 3 min (arrow) with or without rapamycin. FRET ratio = 1.3 (horizontal dotted line) represents maximal Aurora B activity in cells with no INCENP depletion. The experiment was repeated three times with similar results.

The following figure supplement is available for figure 1:

**Figure supplement 1.** Phosphorylation of the chromatin-targeted Aurora B sensor after INbox recruitment to centromeres.

(*Figure 2C*), implying that the partially active Aurora B can activate itself in cis, i.e. intramolecularly (*Figure 2B*).

To reveal the in trans component, we carried out experiments using high concentration of partially active Aurora B, mimicking its clustering at cellular binding sites. At high kinase concentration the chemosensor becomes depleted quickly, so we modified our assay to uncouple the Aurora B autophosphorylation reaction from the activity measurement with the chemosensor (*Figure 2D*). With 4 µM kinase, kinase activity increased strongly with time, and the best-fit curve based only on in cis autoactivation provided a poor fit (*Figure 2D*), confirming the presence of the in trans component. With a computational model combining both reactions (*Figure 2E*), we generated a global fit to experimental curves in *Figure 2A,D* and determined molecular constants for the two-component autoactivation mechanism for Aurora B kinase (*Table 2*, Materials and methods). This model demonstrates that kinase autoactivation in cis dominates over the trans-activation during initial activation at low kinase concentration (*Figure 2—figure supplement 2*, panels H and I).

## A coupled Aurora B kinase-phosphatase system exhibits bistability and hysteresis in silico

Our findings above imply that if Aurora B kinase, phosphatase and ATP are mixed together, two reactions should take place simultaneously: Aurora B autoactivation and its inactivation by phosphatase. We constructed a quantitative model for such a coupled kinase-phosphatase system

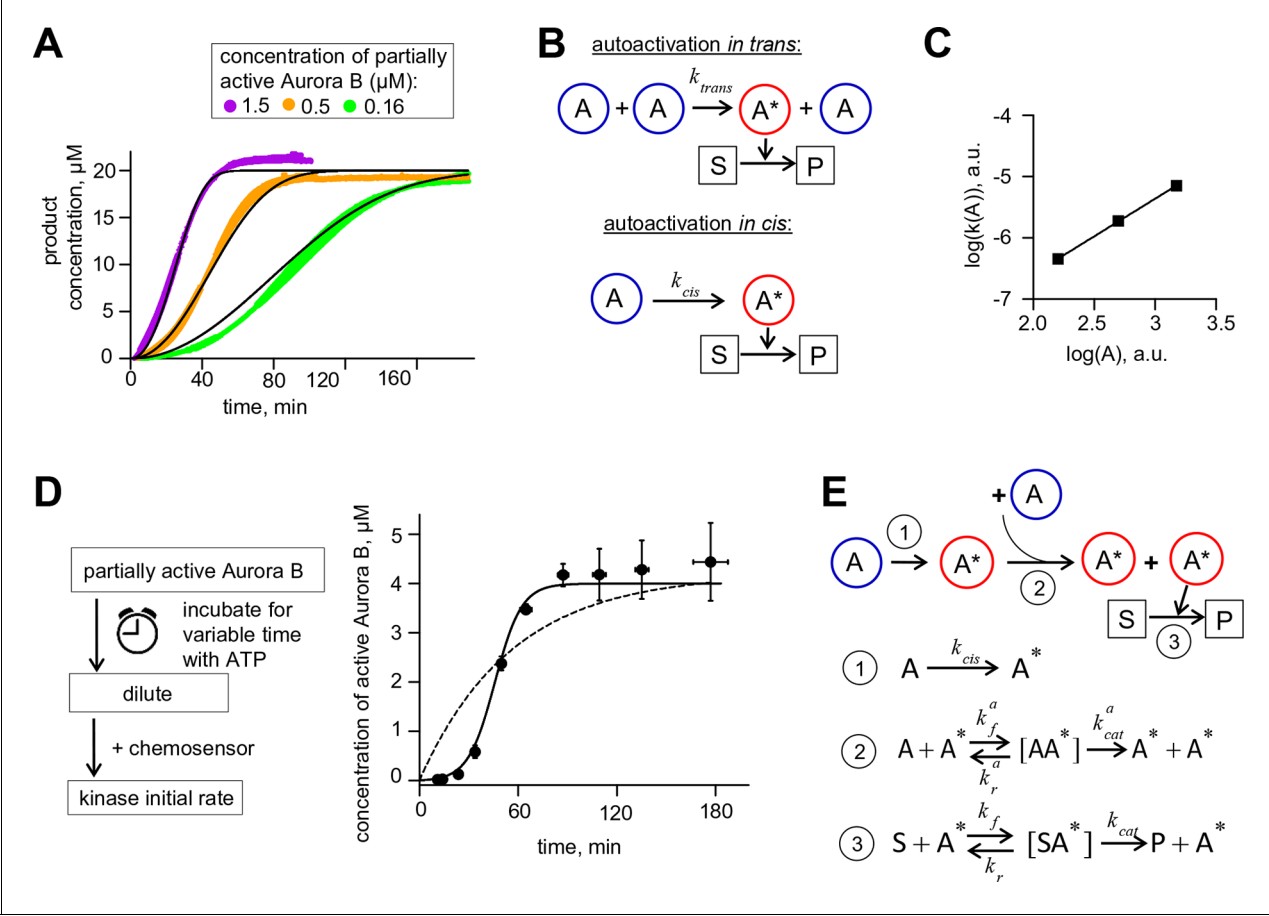

**Figure 2.** Aurora B kinase autoactivation in vitro. (**A**) Phosphorylation of 20 µM chemosensor by the indicated concentrations of partially active Aurora B kinase. Data are averages of N = 2 experiments for each kinase concentration; error bars are SEMs. Black lines are theoretical fittings with the reaction scheme in panel E. (**B**) Molecular scheme for Aurora B autoactivation in trans or in cis. A and A* denote partially active (dephosphorylated) and active kinase; S and P indicate substrate and product (unphosphorylated and phosphorylated chemosensors, respectively). (**C**) Coefficient k for the quadratic phase of chemosensor phosphorylation by partially active Aurora B kinase vs. kinase concentration (**A**) plotted on a log-log scale. Line is linear fit. (**D**) Diagram of the experimental procedure to evaluate Aurora B autoactivation at high kinase concentration (4 µM). Experimental graph on the right shows changes in concentration of active Aurora B, calculated as described in Materials and methods. Data points are mean ± SEM for N≥4 experiments. Solid line is theoretical fitting with the reaction scheme in panel E. Dashed line is theoretical fit using the analytical solution for A*(t) for the reaction scheme with only in cis activation of Aurora B. (**E**) Molecular scheme for the Aurora B kinase two component autoactivation in the presence of chemosensor and the corresponding reactions, see system *equation 2* in Materials and methods. All other symbols are listed in *Tables 1* and *2*.

The following figure supplements are available for figure 2:

**Figure supplement 1.** Bicistronic construct of Aurora B-INbox and its dephosphorylation.

**Figure supplement 2.** Aurora B activity towards chemosensor.

**Figure supplement 3.** Results for Aurora B two sites phosphorylation model.

(*Figure 3A*), which takes into account the determined molecular constants for two-component Aurora B autoactivation and a Michaelis–Menten mechanism for a phosphatase with variable enzymatic constants. Solving the differential equations describing this system in silico (see Materials and methods) reveals that at high kinase concentration three steady-state solutions could coexist (*Figure 3B*). *Figure 3C* shows region of bistability in the parametric plane of Aurora B kinase-phosphatase concentrations. Bistability arises when Aurora B kinase concentration exceeds 4 µM, and further increasing Aurora B concentration broadens the range of permissible phosphatase

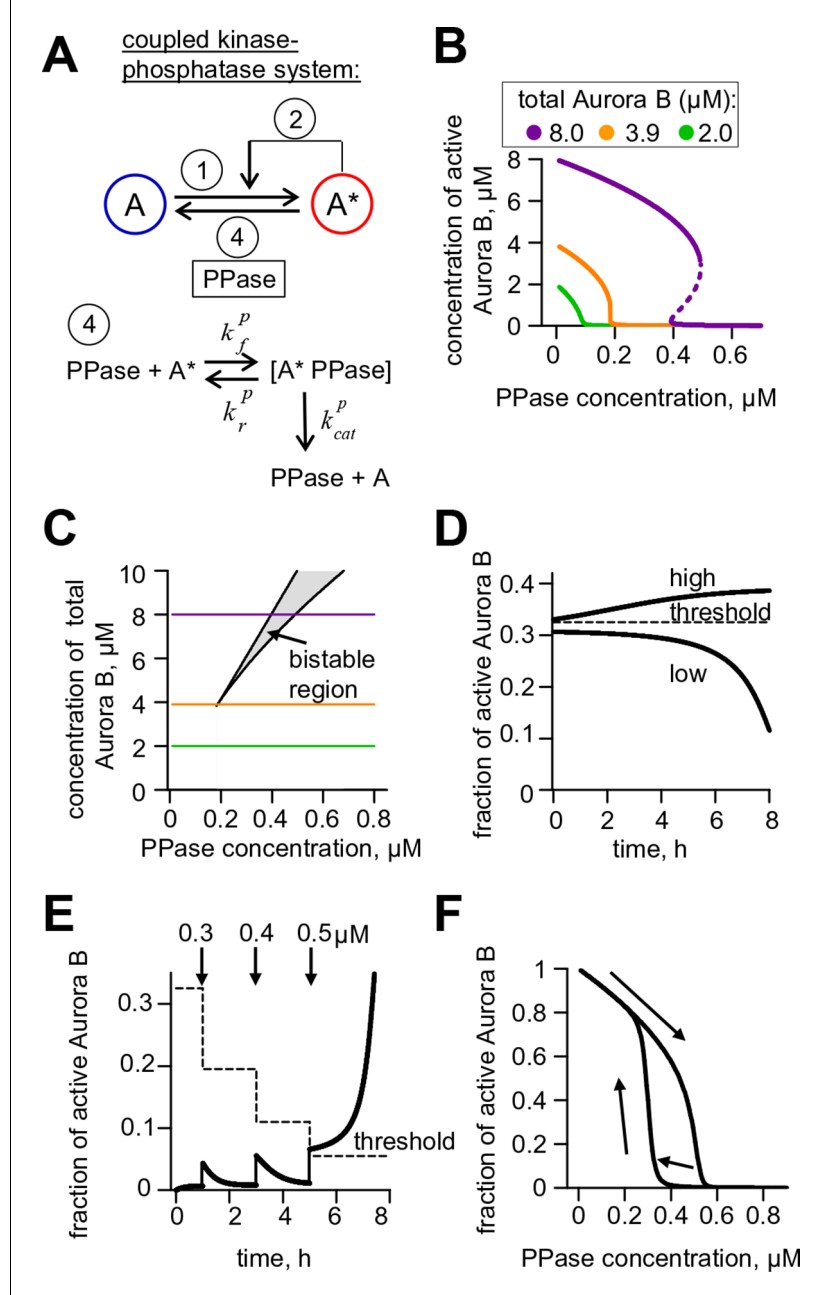

**Figure 3.** Theoretical analysis of the coupled Aurora B kinase-phosphatase system. (A) Molecular scheme for the coupled system and the corresponding reactions. For reactions 1 and 2 see *Figure 2E*; see *Table 1* and *2* for more details. (B) Steady-state solutions for concentration of active Aurora B kinase as a function of phosphatase concentration (*Equation 4* in Materials and methods). For 8 µM Aurora B, three steady states can co-exist: two stable states with high and low activities and one unstable state (dashed line), corresponding to the region of bistability. (C) Bistability region in the parametric plane of phosphatase and total (phosphorylated and not) Aurora B kinase concentrations. In this region the model has two coexisting stable steady-state solutions, while enzymatic concentrations outside this region lead to only one steady state. Colored lines correspond to the solutions shown in panel B for active kinase. (D) Theoretical predictions for the changes in concentration of active Aurora B kinase, plotted as a fraction of total kinase concentration, for two different initial conditions. The initial concentration slightly higher than the threshold (horizontal line) has a steady-state solution with a larger fraction of active kinase (high state). The fraction of active Aurora B kinase declines when its initial concentration is below the threshold (low state). Calculations were done for 8 µM total Aurora B kinase and 0.47 µM phosphatase. (E) Simulation of perturbations to reaction with 0.47 µM phosphatase and 8 µM total Aurora B kinase. Active kinase is added 3

*Figure 3 continued on next page*

*Figure 3 continued*
times as indicated (vertical arrows). The system returns to the steady state with low Aurora B kinase activity until the threshold is exceeded. (**F**) Hysteresis loop in the kinase-phosphatase system with 8 μM kinase. Phosphatase concentration was initially low, so almost all Aurora B kinase was active. As the phosphatase concentration was gradually increased up to 0.8 μM, the steady-state concentration of active Aurora B kinase decreased (top line with downward arrow). Different solutions were obtained when phosphatase concentration was decreased gradually back to 0 μM (lower line with two upward arrows).
The following figure supplement is available for figure 3:

**Figure supplement 1.** Aurora B hysteresis dependency on phosphatase.

concentrations. In this region, a homogeneous mixture of kinase and phosphatase can exist in one of two stable states with different kinase activity, high or low, depending on initial conditions (*Figure 3C*). This prediction is important because, as we will show later, bistable behavior is essential for accurate regulation of Aurora B kinase activity away from sites of high kinase concentration.

As expected for the bistable regime, increasing concentration of active Aurora B above a threshold causes this biochemical system to switch between two states with no intermediate steady-states (*Figure 3D,E*). The model also predicts hysteresis in the region of bistability. Hysteresis becomes evident at intermediate levels of phosphatase concentration (e.g. 0.4 μM in *Figure 3F*), when almost the entire kinase pool can be either phosphorylated (high kinase activity state) or dephosphorylated (low kinase activity state) depending on the prior state of this reaction mixture. Importantly, we find that these non-linear regimes are determined mostly by the parameters of the two component Aurora B autoactivation mechanism, but not by the enzymatic constants of the protein phosphatase (*Figure 3—figure supplement 1*; see Materials and methods). Thus, Aurora B kinase, coupled with an inactivating phosphatase, is predicted to exhibit robust hysteresis and bistability.

## Aurora B kinase-phosphatase bistability and hysteresis observed in vitro are in a quantitative agreement with theoretical predictions

Using the reconstituted in vitro system, we next designed an experiment to test the prediction of our theoretical model that at high Aurora B concentration the same mixture of kinase and phosphatase will result in different degrees of Aurora B activity depending on the initial conditions. We combined Aurora B kinase (8 μM), ATP (4 mM) and variable concentrations of λ phage phosphatase, such that both Aurora B activation and inactivation could take place simultaneously. Importantly, these reactions were carried out for two different initial conditions: using either the active kinase or active kinase pretreated with phosphatase (*Figure 4A*). All other reactants in these two mixtures were adjusted to achieve the same final concentrations for all components, including the phosphatase. The progress of these reactions was followed by taking samples at the indicated times; phosphatase inhibitor was then added to stop kinase dephosphorylation and kinase activity was measured via the initial rate of chemosensor phosphorylation. As expected, at low phosphatase concentration (0.25 μM) the partially active kinase gradually activated itself, reaching a steady-state with high activity, while the active kinase slightly lost its activity (*Figure 4B*, top graph). At high phosphatase concentration (0.5 μM, bottom graph) the active kinase was overpowered by the phosphatase and became gradually inactivated, reaching a level close to the fully dephosphorylated kinase. Importantly, the model accurately predicted the behavior for these two reactions and when intermediate phosphatase concentration was used (lines in *Figure 4B*). At 0.45 μM phosphatase, the kinase that was initially active robustly retained its high state (red data points, *Figure 4B*, middle graph), while the activity remained low for the kinase that was initially in the low state (blue data points, *Figure 4B* middle graph). These outcomes demonstrate bistability because in both of these enzyme mixtures the final concentrations of all components were identical and the reactions were allowed to proceed long enough to reach the steady-states (120 min). Similar experiments were carried out for additional phosphatase concentrations, and the steady-state levels of active Aurora B were obtained by averaging measurements for ≥60 min incubation times. These data, plotted as a function of phosphatase concentration in *Figure 4C*, define the bistable region for the homogeneous system in vitro, in quantitative agreement with model predictions. In this range of concentrations, the coupled

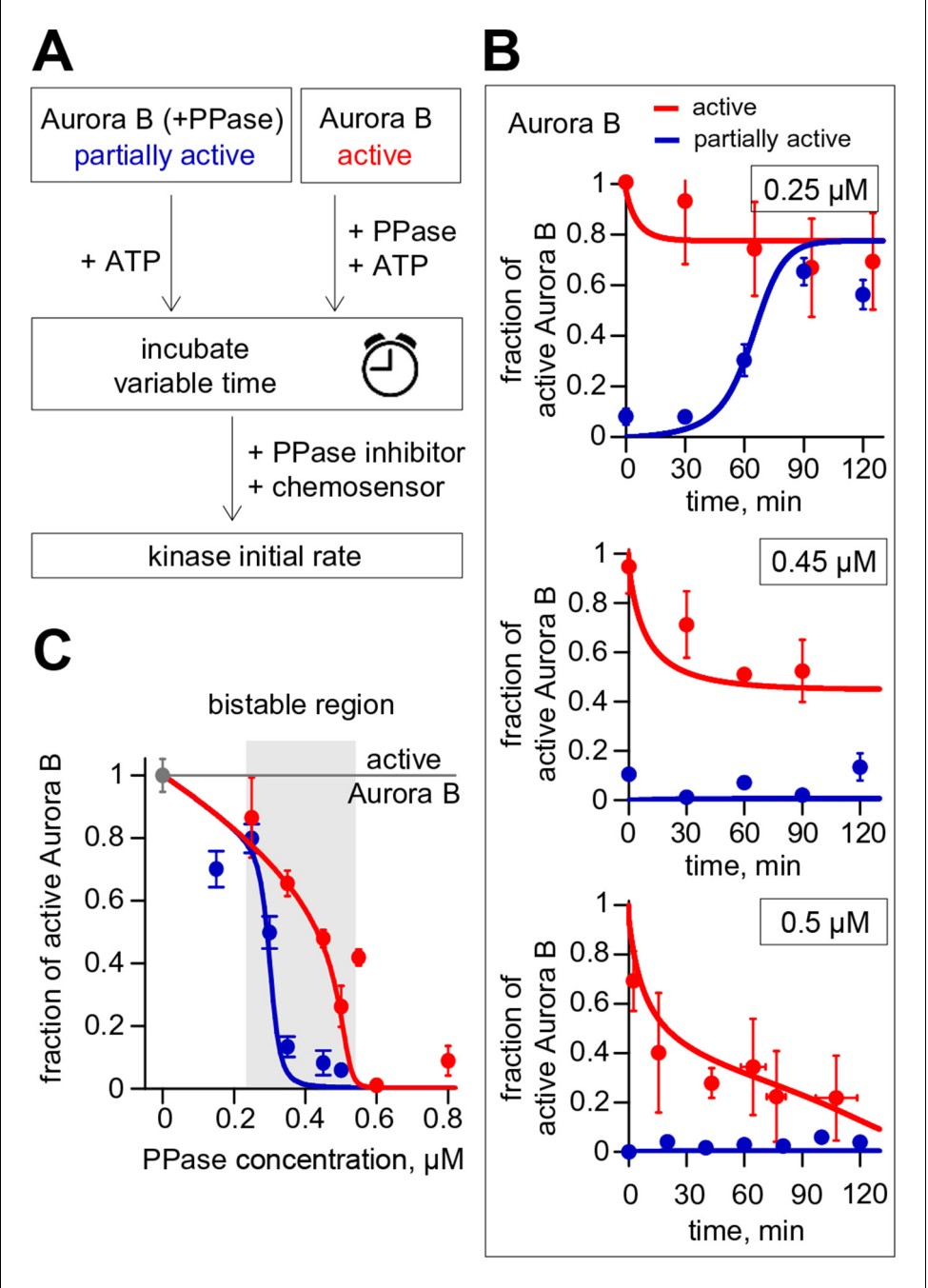

**Figure 4.** Reconstitution of the coupled Aurora B kinase-phosphatase system in vitro. (**A**) Diagram of the experimental procedure to study bistability and hysteresis. Active kinase was preincubated with phosphatase (PPase) in the absence of ATP to generate partially active kinase, and ATP was added at time = 0 ('initially low' experiment). In a parallel experiment, the same reagents were used but active kinase, phosphatase and ATP were mixed together at time = 0 ('initially high' experiment). Samples were taken to analyze kinase activity until the corresponding steady states were reached. (**B**) Experimental results (dots) for changes in kinase activity vs. incubation time for 8 µM kinase and 0.25, 0.45 or 0.5 µM phosphatase, as indicated. Each point shows mean ± SEM (N≥2) for experiments with active (red) or partially active (blue) Aurora B kinase. (**C**) Fraction of active kinase at steady state as a function of phosphatase concentration. Points are mean ± SEM for N≥4 independent experiments. These data are in close agreement with the model built using our experimentally determined kinetic constants (solid lines in panels B and C).

kinase-phosphatase system exhibits hysteresis, with different activity levels observed depending on the initial conditions.

## Evidence for bistability and hysteresis of Aurora B kinase activity in mitotic cells

If the biochemically simplified coupled kinase-phosphatase system in our in vitro experiment represents the behavior in the more complex in vivo setting, we predict that complex non-linear regimes should also be observed in mitotic cells, where Aurora B kinase is highly concentrated at the sites of its localization and various cellular phosphatases, such as PP1 (*Trinkle-Mulcahy et al., 2003*), may inactivate it by dephosphorylation. First, we tested whether the endogenous Aurora B–phosphatase system can exhibit bistability, using a FRET-based phosphorylation sensor targeted to centromeres by fusion to CENP-B (*Fuller et al., 2008*). If the system is bistable, this sensor should reveal that Aurora B exists in either a high or low activity state and that these states can co-exist under the same experimental conditions. We manipulated Aurora B activity by incubating mitotic cells with varying concentrations of its specific inhibitor, ZM447439. Cells were imaged live, and the average FRET ratio was calculated for each individual cell, representing the overall phosphorylation state of that cell. Analysis of a population of cells expressing the centromere-targeted sensor shows that the distribution of phosphorylation states is clearly bimodal, with distinct high and low FRET states (*Figure 5A*). In the absence of inhibitor, all cells are in the high phosphorylation (low FRET ratio) state, as expected. As the inhibitor concentration increases, the distribution shifts so that a greater fraction of cells is in the low phosphorylation state, but intermediate phosphorylation states are rare. Importantly, for some intermediate concentrations of Aurora B inhibitor, both peaks are observed in the same cell population, likely because individual cells differ slightly in their parameters, for example in membrane permeability to kinase inhibitor. Similar results were obtained for the sensor targeted to chromatin by fusion with histone H2B (*Fuller et al., 2008*). Here, cells were blocked in mitosis with either monastrol or nocodazole (*Figure 5B*), indicating that these results do not depend on which sensor is used or how cells are arrested.

To further test the bistable kinase-phosphatase system in vivo, we asked whether the prior history of Aurora B kinase activation affects the level of Aurora B activity in mitotic cells. We designed an experiment to manipulate Aurora B activity in a similar manner as in our in vitro experiments, in which hysteresis was observed. Cells were first incubated with low (0 µM) or high (1.5 µM) concentration of the Aurora B inhibitor to establish two different initial conditions of either high or low kinase activity, respectively. From these initial conditions, the inhibitor concentration was switched following one of four protocols: low to high (0 to 1.5 µm) or high to low (1.5 to 0 µM), as experimental controls, and low to intermediate (0 to 0.6 µM) or high to intermediate (1.5 to 0.6 µM) to reach identical final conditions. If hysteresis is present, cells that end up at the same intermediate inhibitor concentration will show different phosphorylation levels depending on their past history, i.e., whether they we preincubated with initially high or low inhibitor concentration. Cells were then imaged live to track changes in phosphorylation of the chromatin targeted FRET sensor. Switching from low to high inhibitor concentration led to kinase inhibition and sensor dephosphorylation, and conversely switching from high to low led to kinase activation and sensor phosphorylation, as expected. When the inhibitor concentration was switched to the intermediate level, however, kinase activity remained in the initial state in these mitotic cells: high if the initial condition was low inhibitor (0 to 0.6 µM) and low activity if the initial condition was high inhibitor (1.5 to 0.6 µM) (*Figure 5C,D*). In addition, we found that cellular localizations of Aurora B and PP1γ phosphatase were not affected by treatment with this inhibitor (*Figure 5—figure supplement 1*), consistent with a previous report for Aurora B localization (*Ditchfield et al., 2003*). Together, these results strongly indicate that bistability and hysteresis of Aurora B phosphorylation in mitotic cells are driven by the intrinsic properties of Aurora B kinase coupled with the inactivating phosphatase(s).

## A quantitative model links non-linear behavior of the coupled Aurora B kinase-phosphatase system with bistability and hysteresis observed in cells

To gain insight into the physiological significance of the bistability of the Aurora B kinase-phosphatase system, we built a spatial model of Aurora B kinase activity in cells. This model simplifies or

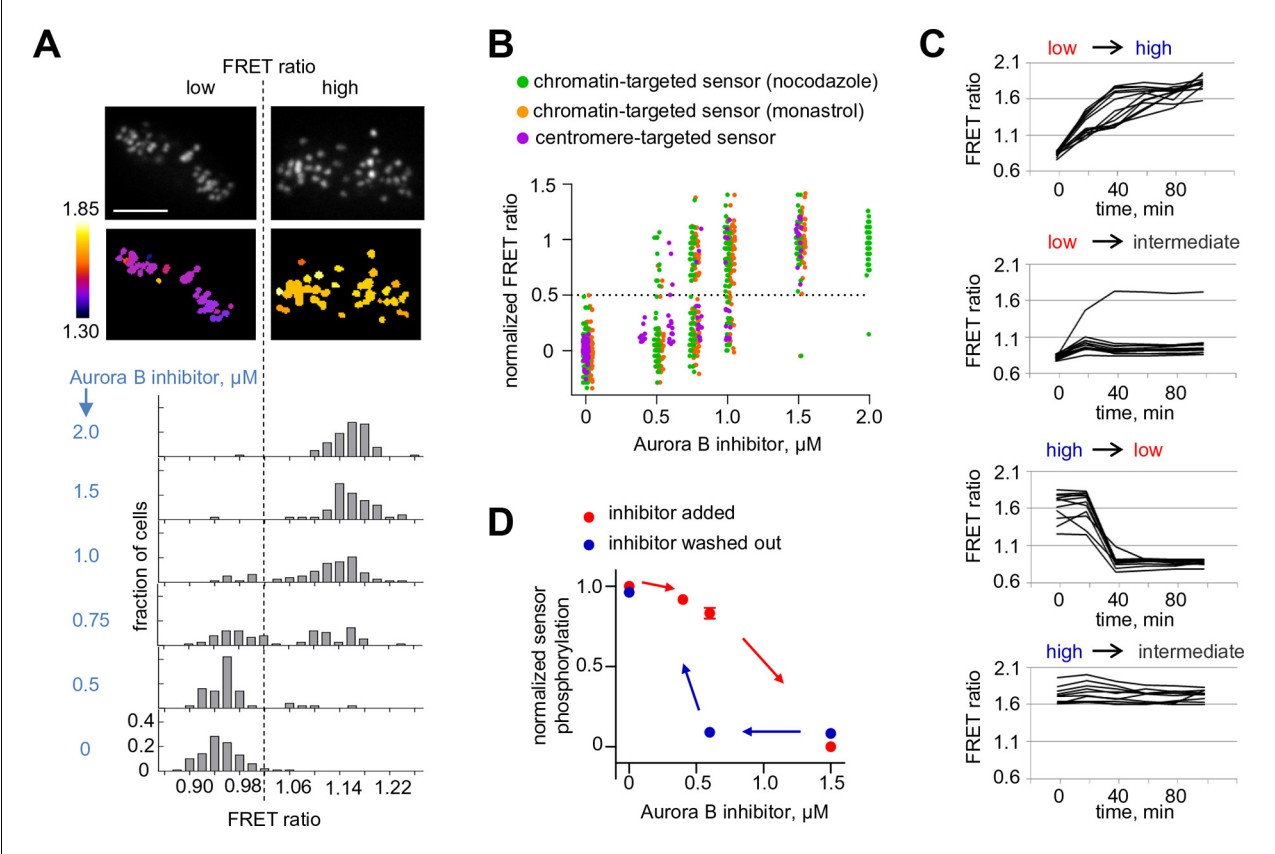

**Figure 5.** Bistability and hysteresis of Aurora B kinase in dividing cells. (**A**) Cells expressing the centromere-targeted Aurora B sensor were arrested in mitosis with the proteasome inhibitor MG132 and incubated with various concentrations of the Aurora B inhibitor ZM447439, then imaged live. Images show representative cells at 0.75 µM of Aurora B inhibitor: top images CFP emission, bottom images YFP/CFP emission ratio. Histograms below show fraction of cells with the indicated FRET ratio (average YFP/CFP over a single cell), N > 50 cells for each Aurora B inhibitor concentration. (**B**) Data from (**A**) are plotted together with similar experiments for cells expressing the chromatin-targeted sensor, arrested in mitosis with either nocodazole or monastrol and incubated with different ZM447439 concentrations. Each data point represents the FRET ratio for one cell normalized as described in Materials and Methods. (**C**) HeLa cells expressing the chromatin-targeted Aurora B sensor were arrested in mitosis with nocodazole and treated with either 0 or 1.5 µM ZM447439 for 100 min. Then, cells were imaged live and ZM447439 concentration was changed as indicated at t = 0. Results of a single experiment are plotted with each line representing an individual cell; 'low', 'intermediate' and 'high' correspond to ZM447439 concentrations 0, 0.6 and 1.5 µM, respectively. (**D**) Normalized steady-state sensor phosphorylation as a function of ZM447439 concentration. Each data point (mean ± SEM) is calculated from the average of final FRET ratios for cells imaged as in panel (**C**) (see Materials and methods). When the final FRET ratio is at a minimum (as in 0 µM inhibitor), the normalized sensor phosphorylation is maximal because phosphorylation decreases FRET in this biosensor. Data were averaged over two independent experiments, N>9 cells per condition in each experiment.

The following figure supplement is available for figure 5:

**Figure supplement 1.** Aurora B and PP1γ localizations are not affected by Aurora B inhibition.

leaves out many mitotic features while focusing on molecular processes that are essential for Aurora B kinase activity in cells. Specifically, we used deconvolved intensity profiles for Aurora B localization at the centromere and along chromosome arms to define the spatial distribution of Aurora B binding sites on chromatin (*Figure 6—figure supplement 1*). Peak Aurora B concentration at the centromere is estimated at 10 µM, dropping down to 1.5 µM along chromosome arms and 1–2 µM in the kinetochore area (*Figure 6A*). In the model soluble kinase molecules bind and unbind these sites dynamically to achieve the steady-state fractions of the bound and diffusing soluble kinase pools of 75% and 25%, respectively (*Mahen et al., 2014*). Soluble kinase in the model behaves identically to our in vitro experiments, activating itself via the two component mechanism with the kinetic constants listed in *Table 2*. The activity of chromatin-bound Aurora B is not known, but the elongated flexible

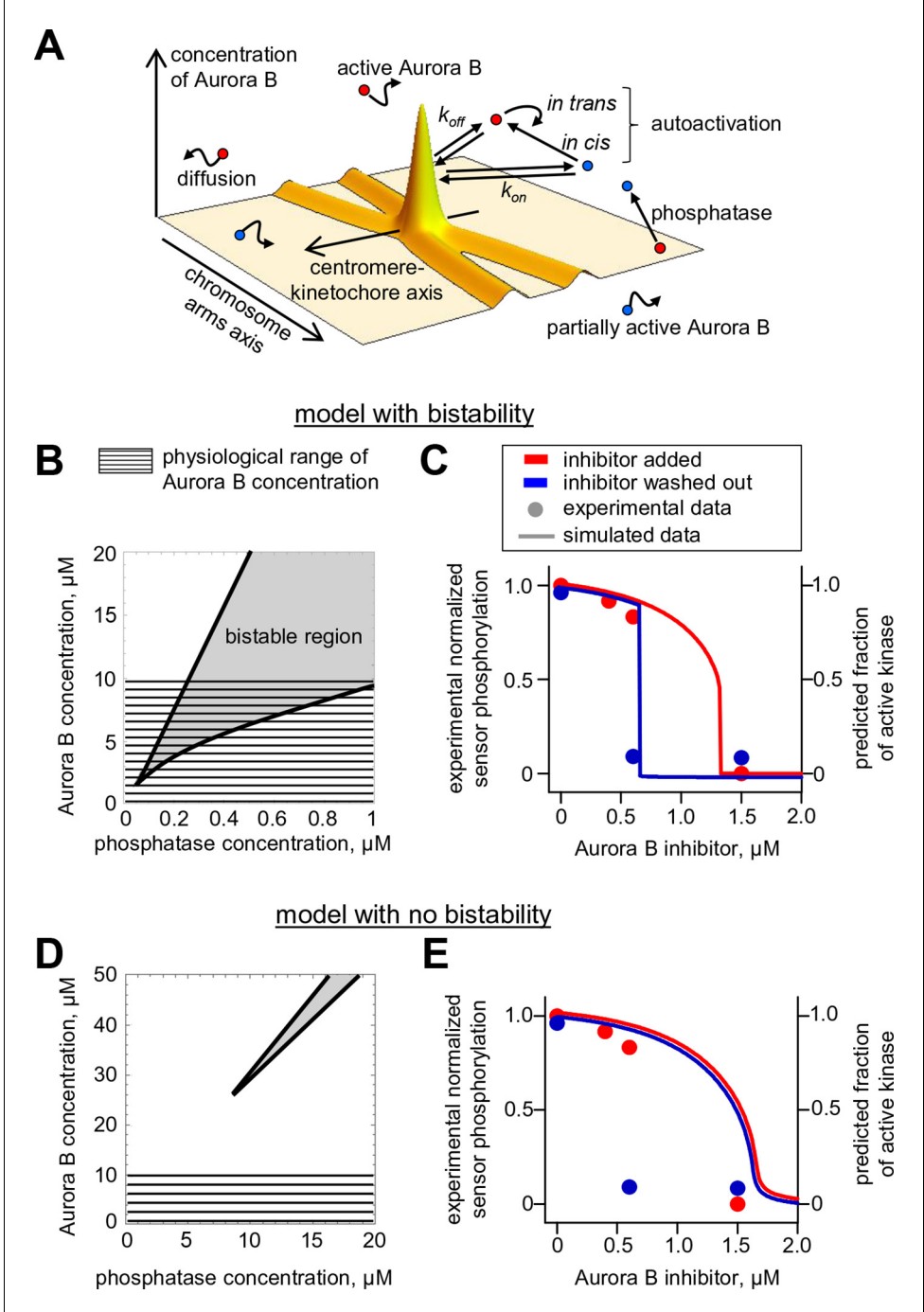

**Figure 6.** Spatial model of Aurora B activity in the cell. (A) Schematics of the essential features of our spatial model, see Materials and methods for details. (B) Parametric plane of phosphatase and total Aurora B kinase concentrations analogous to the plot in *Figure 3C* but calculated using the spatial model. Grey area shows region of bistability; $K^P_M = 0.16$ µM. (C) Simulated (right axis) and experimental (left axis) results for the hysteresis experiment in cells (data points reproduced from *Figure 5D*). Calculations are for $K^P_M = 0.16$ µM, phosphatase concentration 0.1 µM. (D,E) These plots are analogous to those in panels (B,C), but they were calculated for $k_{cis} = 7.3 \cdot 10^{-4}$ s$^{-1}$, all other model parameters were not changed. With this autoactivation constant, the model predicts no bistability in the physiological range of kinase concentrations (D), and, the kinase activity vs. inhibitor concentration curve does not depend on the system's history (E), blue and red curves are slightly offset for clarity).

The following figure supplement is available for figure 6:

**Figure supplement 1.** Theoretical model for spatial regulation of Aurora B kinase phosphorylation.

structure of the INCENP subunit is thought to permit some mobility for the tethered Aurora B kinase (*Krenn and Musacchio, 2015*; *Samejima et al., 2015*). We therefore assume that chromatin bound Aurora B kinase can interact with soluble Aurora B molecules freely (with the same kinetic constants as in *Table 2*, but phosphorylation in trans between the chromatin-bound molecules is limited (see Materials and methods). Both bound and soluble Aurora B molecules can be inactivated by a phosphatase, which for simplicity is assumed to be soluble and diffusive. This reaction-diffusion system was described with partial differential equations (*Equation 6* in Materials and methods) and solved numerically.

With this analytical framework we first tested if this model could reproduce the bistability and hysteresis observed for the overall state of Aurora B kinase in cell experiments. Consistent with our theoretical analysis of the homogeneous system, bistability was predicted for a range of phosphatase concentrations, overlapping with the physiological kinase concentrations (*Figure 6B*). Aurora B inhibition was then simulated using the published relationship between ZM447439 concentration and Aurora B activity (*Ditchfield et al., 2003*). The steady-state Aurora B activity was examined starting from two different initial conditions: when all cellular kinase had enzymatic activity of the fully active kinase or it was inactive. The inhibitor concentration was varied in 10-nM steps, and the spatial distribution of Aurora B kinase activity was calculated and averaged to represent the overall fraction of active kinase for each initial condition and inhibitor concentration (*Figure 6C*, lines). As in the homogeneous system (*Figure 3—figure supplement 1*), the $K^P_M$ value for phosphatase affected the exact shape and position of this theoretical hysteresis plot. The enzymatic constant for phosphatase acting on Aurora B kinase in cells is not known, but $K^P_M$ = 0.16 µM provided an excellent match with experimental measurements in cells (*Figure 6C*). Thus, Aurora B hysteresis in mitotic cells can be reproduced using the molecular and biochemical features which form the basis for our model and reasonable values of model parameters.

To examine whether bistability of the coupled kinase-phosphatase system was essential for matching the experimental data, we modified our model by changing only one parameter $k_{cis}$, which characterizes Aurora B kinase autophosphorylation in cis. Importantly, all other model features and the values of all other parameters were unchanged, such that Aurora B autoactivation and its inhibition by phosphatase were still present. With this modification, the bistable region could only be observed at much higher kinase and phosphatase concentrations, while bistability in the range of physiological Aurora B concentrations was lost (*Figure 6D*). When this modified model was used to mimic the ZM447439 inhibition experiment, it predicted a reasonably good match to the gradual decrease in Aurora B activity in experiments with increasing inhibitor concentration. However, when calculations were done starting from the inactive kinase and the inhibitor was 'washed out', the model prediction did not change, indicating a lack of hysteresis (*Figure 6E*). Thus, bistability of the underlying biochemical pathways is required to explain hysteresis that we detected with the Aurora B phosphorylation sensor in cells.

## Bistability underlies spatial patterns of Aurora B phosphorylation in mitotic cells

Next, we investigated model predictions for the regulation of Aurora B phosphorylation of substrates located remotely from centromeric sites of kinase enrichment. Previous experiments using cells arrested in mitosis found gradients of Aurora B phosphorylation spreading from centromeres, along chromosome arms, after Aurora B was inhibited with ZM447439 and then the inhibitor was washed out (*Wang et al., 2011*) (*Figure 7A*). Analogous images were obtained in this work after inducible clustering of Aurora B at the centromere (*Figure 1B* and *Figure 1—figure supplement 1B*), emphasizing that these large scale phosphorylation patterns are triggered by Aurora B localization and activation at the centromere. We modeled the kinase inhibitor washout experiment to determine the spatiotemporal distribution of activated Aurora B kinase, then calculated the resulting phosphorylation patterns for a chromatin-bound substrate (see Materials and methods). Consistent with the in vivo experiment, the model exhibited spatially non-uniform large-scale distributions with phosphorylation high at the centromere and gradually decreasing along chromosome arms (*Figure 7A*). In the model, and in cells, these gradients are transient, as Aurora B signal propagates from the centromere, eventually leading to uniformly high Aurora B phosphorylation of all chromatin bound substrates (*Figure 7B*). This spreading appeared similar to a trigger wave, a hallmark feature of a bistable medium (*Kapral and Showalter, 1995*). Importantly, self-sustained trigger waves

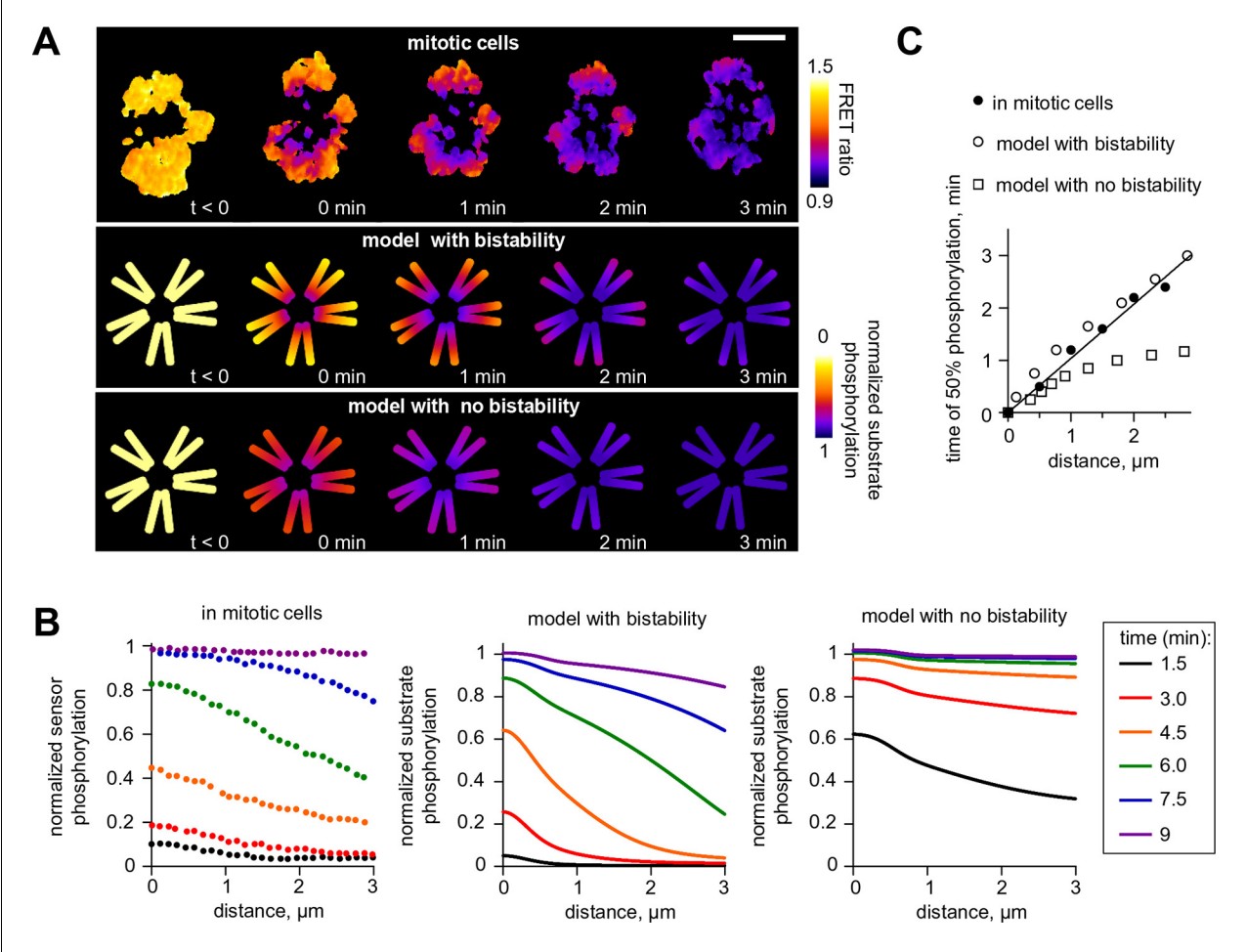

**Figure 7.** Wave propagation of Aurora B activity. (**A**) Color-coded plots showing spatial patterns of Aurora B phosphorylation. Top row: HeLa cell expressing the chromatin-targeted FRET sensor and arrested with monastrol is shown before (t<0) and after Aurora B inhibitor washout. Time 0 min corresponds to FRET signal reaching half of its maximum level at the centromere. Lower FRET signal corresponds to higher sensor phosphorylation. Other two rows: color-coded substrate phosphorylation calculated in the models with and without bistability. Scale bar, 5 μm. (**B**) Profiles of average substrate phosphorylation along chromosome arms in cells observed at different time after inhibitor washout and analogous model predictions. Signals were normalized to maximum level of substrate phosphorylation. (**C**) Time of 50% sensor phosphorylation as a function of distance along chromosome arms. Closed symbols: experimental data with a linear fit. Open symbols correspond to model solutions with and without bistability.

propagate at a constant speed, which discriminates them from other mechanisms of signal propagation in systems with diffusion. Indeed, the predicted plot for the timing of Aurora B activation as a function of distance from the centromere was linear, implying a constant rate of spreading (*Figure 7C*). To test this model prediction we plotted the time of phosphorylation as a function of distance from centromeres for the chromatin bound FRET sensor, from experiments in which ZM447439 was washed out. This dependency was also linear, strongly suggesting that phosphorylation along chromatin propagates as a trigger wave (*Figure 7C*). As expected, the model with no bistability in the concentration range of chromatin bound Aurora B predicted very different kinetics of phosphorylation spreading with a non-linear rate (*Figure 7A–C*).

Finally, we used our model to seek new insights into the spatial distribution of Aurora B kinase activity at kinetochores, where phosphorylation decreases from prometaphase to metaphase. Previous measurements in metaphase using Aurora B phosphorylation sensors targeted to different molecular locations at kinetochores revealed different phosphorylation levels at sites separated by only 10s of nm, indicating a sharp gradient of Aurora B activity (*Welburn et al., 2010*; *Suzuki et al., 2014*). With our model, we calculated the fraction of activated kinase as a function of distance from

the centroid (midway between the sister kinetochores) for the unstretched centromere, corresponding to the microtubule-free kinetochores in prometaphase. Almost the entire centromere-bound pool of prometaphase Aurora B kinase is predicted to be active, with the fraction of active kinase decreasing slightly at the kinetochore (bounded by CENP-A and Ndc80) (*Figure 8A*). We then stretched this mechano-biochemical system to mimic the ~2-fold increase in distance between sister kinetochores seen in metaphase HeLa cells (*Wan et al., 2009*). In stretched chromatin the distance between Aurora B binding sites increased correspondingly, as indicated with the white mesh in *Figure 8*. As a result, the local concentration of chromatin-bound kinase decreased, and a region of bistability emerged at the kinetochore, hundreds of nm away from the centroid (*Figure 8—figure supplement 1*). As we have shown earlier, the bistable kinase-phosphatase system exhibits a highly nonlinear behavior. In the chromatin meshwork, these threshold-dependent reactions created a stable and steep gradient of Aurora B kinase activity. In contrast, in the model with no bistability, stretching induced a much more gradual gradient of Aurora B activity, reflecting a gradual decrease in density of centromere-bound Aurora B kinase (*Figure 8B,D*).

## Discussion

Our findings address the long-standing question of how Aurora B phosphorylates substrates at a distance from its major sites of localization. First, we use a rapamycin-induced targeting system to examine the immediate effects of concentrating Aurora B at centromeres. This approach represents an advance over previous manipulations of Aurora B localization, such as depleting CPC components, comparing different mutant forms of INCENP, or inhibiting mitotic kinases that control CPC localization (*Vader et al., 2006*; *Tsukahara et al., 2010*; *Wang et al., 2010*; *2011*; *Yamagishi et al., 2010*). In the experiments reported here, Aurora B localization is controlled by rapamycin addition,

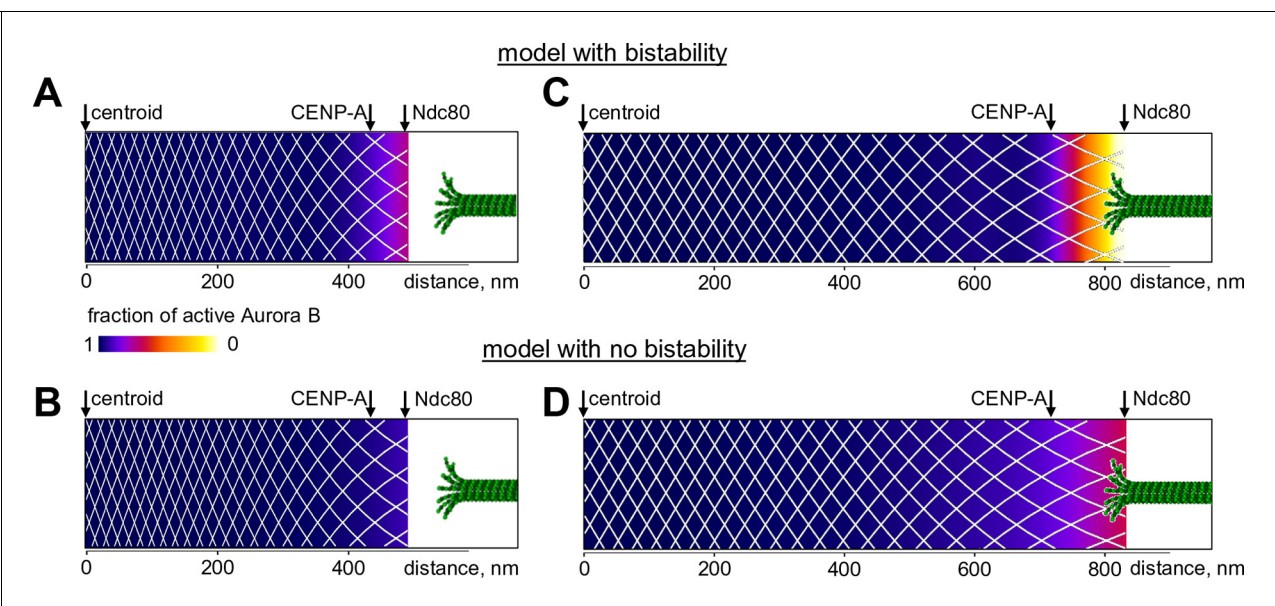

**Figure 8.** Predicted gradient of Aurora B kinase activity at kinetochores during prometaphase (left) and metaphase (right). Color-coded plots of the profile of Aurora B kinase activity along the axis connecting the centromere centroid (midway between the sister kinetochores) and the outer kinetochore. Arrow for Ndc80 corresponds to the location of the N-terminus of Hec1 (*Wan et al., 2009*). Density of the white mesh indicates concentration of Aurora B kinase; local Aurora B concentration is lower when mesh holes are larger. (A) and (B) show model predictions for prometaphase kinetochores that are not under tension (smaller centroid to Ndc80 distance). In metaphase (C and D) this distance increases due to forces generated by the end-on attached kinetochore microtubules. In the model without bistability (B–D), the fraction of active Aurora B kinase simply reflects the total Aurora B kinase concentration.

The following figure supplement is available for figure 8:

**Figure supplement 1.** Quantification of Aurora B activity gradient during mitosis.

while keeping other components of the system constant, so indirect effects from these manipulations are less likely. Importantly, in the absence of rapamycin, the cytosolic Aurora B/INbox protein complex shows little activity toward chromatin-localized targets. However, recruiting the same complex to centromeric binding sites leads to phosphorylation of the chromatin-localized probe within minutes, demonstrating that the highly-concentrated centromeric pool of Aurora B is essential for phosphorylation at other cellular locations (*Figure 1B*). This result from mitotic cells suggests that specialized mechanisms enable long-range regulation of Aurora B kinase activity in mitosis.

The theory of complex non-linear systems suggests a plausible molecular explanation for this phenomenon, since certain feedback-controlled reactions are known to lead to formation of a self-sustained source of active components and establishment of well-controlled spatial activity patterns (*Kapral and Showalter, 1995*). Testing such biochemical models requires knowledge of the underlying feedbacks and specific enzymatic constants and parameters values. In this work we build a quantitative foundation for such a mechanism using a reconstituted system with purified components. First, our work defines a quantitative biochemical mechanism for Aurora B autoactivation (*Table 2*). Previous experiments have suggested that Aurora B is activated in trans (*Bishop and Schumacher, 2002*; *Honda et al., 2003*; *Sessa et al., 2005*; *Kelly et al., 2007*). We confirm these initial findings, but we also find that only the active, already phosphorylated Aurora B kinase can activate in trans. The analogous reaction by the unphosphorylated kinase is not as productive and is carried out in cis. This newly revealed cis component dominates at initial stages of Aurora B kinase activation, when the majority of kinase molecules are still unphosphorylated. The cis step may reflect autophosphorylation of the activation loop, as shown for Aurora A (*Dodson et al., 2013*), while the TSS motif of INCENP is phosphorylated in trans. The significance of the in cis reaction is not fully understood, but we show that its rate has a large impact on the bistable region of the Aurora B-phosphatase system, as discussed below.

Second, with a mathematical model we show that the kinetic constants we have defined for Aurora B kinase autoactivation can lead to non-linear behavior, bistability and hysteresis, when Aurora B is coupled with a phosphatase (*Figure 4*). Positive feedback in this system is provided by the two-component (cis and trans) autoactivation of Aurora B, while protein phosphatase inactivates the kinase by dephosphorylation. Importantly, we were able to observe this non-trivial behavior in vitro using purified Aurora B kinase and λ phosphatase, which to the best of our knowledge is the first reconstitution of this kind for any kinase-phosphatase pair. The experimental results in vitro are in quantitative agreement with model predictions (*Figure 4*), implying that we have reached a deep understanding of these phenomena.

Importantly, our theoretical analyses predicted that bistability depends strongly on the Aurora B kinase autoactivation mechanism, while kinetic constants for the dephosphorylation reaction have much less impact (*Figure 3—figure supplement 1*). Thus, although our simplified reconstitution in vitro used the non-physiological λ phosphatase, the model suggested that these nonlinear regimes could exist in a physiological cell context, where Aurora B kinase is coupled with its native phosphatase partner(s). Indeed, we were able to recreate bistability and hysteresis for Aurora B substrate phosphorylation in live mitotic cells. Our cellular experiments relied on sensors at different locations (centromere or chromatin) and used three methods to synchronize cells (monastrol, nocodazole, or the proteasome inhibitor MG312). Because these different experimental tools led to consistent results, our findings in cells likely reflect the same basic non-linear mechanisms that we recapitulated in vitro. We therefore explain the distinct phosphorylation states in mitotic cells as arising from the threshold-dependent autoactivation of the highly concentrated pool of centromere-localized Aurora B, propagated at long distance via the chromatin-bound and cytosolic pools of Aurora B. As in vitro, the high kinase activity state in cells can be sustained over a range of input signals, which in cells were generated using different concentrations of a specific Aurora B inhibitor. As the inhibitor concentration was increased, the system switched to the low kinase activity state (*Figure 5A,B*), just as happened in vitro and in our model when the threshold was crossed. Moreover, these 'high or low' kinase activity states persisted in populations of mitotic cells under the same conditions, depending on the initial state (*Figure 5C,D*). The consistency between our findings in vitro and in vivo indicates that we have captured key features of the Aurora B-phosphatase system that underlie cellular behaviors.

This conclusion is also supported by our ability to describe the in vivo results for bistability and hysteresis using a spatial model of Aurora B kinase activity in mitotic cells. Since the cellular

environment for Aurora B regulation is complex and many constants for the underlying biochemical and molecular reactions in cells are not known, it is currently not possible to provide a detailed quantitative description of Aurora B phosphorylation in cells. However, using reasonable assumptions and values for unknown model parameters, we were able to recapitulate our findings of hysteresis in live cells (*Figure 6*). Moreover, the model made a strong prediction for the propagation of self-sustained trigger wayves of kinase phosphorylation. Long-range propagation of Aurora B phosphorylation has been observed previously (*Wang et al., 2011*), but it was thought to be caused by simple diffusion of activated Aurora B released from the sites of concentration. We quantified these waves and found that they propagate at constant speed (*Figure 7*), strongly implying that they are sustained by a more complex reaction-diffusion mechanism in a bistable system. Additionally, we demonstrate that a highly similar reaction-diffusion model, which also includes kinase autoactivation coupled with inactivating phosphatase but lacks bistability in the range of physiological Aurora B concentrations, fails to reproduce the trigger waves and other results in cells. We conclude that bistability of the coupled kinase-phosphatase system is an essential feature of Aurora B kinase regulation in cells.

Different non-linear mechanisms operating in excitable media have been shown to play important roles in developmental biology and cell division (*Turing, 1952*; *Caudron et al., 2005*; *Maini et al., 2006*; *Karsenti, 2008*; *Chang and Ferrell, 2013*), the cardiovascular system and blood clotting (*Lobanova and Ataullakhanov, 2003*; *Sharma et al., 2009*), kinase signaling (*Kholodenko, 2009*) and intracellular patterning and size control (*Meyers et al., 2006*; *Fischer-Friedrich et al., 2010*; *Hachet et al., 2011*; *Subramanian et al., 2015*). Based on our findings, we propose that bistability of a coupled Aurora B-phosphatase system enables formation of an excitable medium, in which a source of localized active kinase can trigger complex spatial patterns, orchestrating Aurora B phosphorylation in a time and location-dependent manner. Specifically, our work offers a plausible physico-chemical mechanism to explain long-distance regulation of phosphorylation at the mitotic kinetochore in response to tension. We view the centromeric chromatin as a mechanical medium that is capable of sustaining biochemical reactions via a spatially non-uniform distribution of Aurora B kinase. Activity of this kinase in different chromatin areas depends on both the local concentration of chromatin-tethered Aurora B molecules and on the activity of the soluble Aurora B pool. Importantly, our work clearly shows that the level of Aurora B activity in each local area also depends on Aurora B activity in more distant locations of this mechano-biochemical medium. This is because in areas with higher kinase concentration, such as the inner centromere, kinase is strongly activated due to the two component autoactivation mechanism. This activity then propagates to more distant areas with lower kinase concentration via phosphorylation in trans by neighboring chromatin-tethered kinase molecules and via their cross-talk with diffusing soluble kinase. Since the concentration of chromatin-bound kinase decreases from the inner centromere to the outer kinetochore, this reaction-diffusion system can establish a gradient of kinase activity even if the underlying biochemical pathways are not bistable (*Figure 8* and *Figure 8—figure supplement 1*). However, we demonstrate that when the bistable coupled kinase-phosphatase system is incorporated into this stretchable mechanical matrix, the resulting phosphorylation gradients can be much steeper. Moreover, the steep part of the gradient arises only upon kinetochore stretching and coincides with the outer kinetochore area, where the concentration range for Aurora B kinase causes its bistability. Thus, bistability affords versatile control of the position and steepness of the resulting gradient, which is thought to be essential for regulation of kinetochore-microtubule interactions (*Funabiki and Wynne, 2013*; *Krenn and Musacchio, 2015*).

Bistability is also likely to play an important role in establishing a gradient of Aurora B activity around the spindle midzone in anaphase, though the mechanistic details may be different and need to be examined separately. In addition to the bistable system described here, other mechanisms may also regulate spatial patterns of Aurora B activity, such as changes in Aurora B enrichment at centromeres as chromosomes align (*Salimian et al., 2011*) and localized phosphatase activity at different cellular locations, such as kinetochores or centromeres or on chromatin (*Trinkle-Mulcahy et al., 2003*; *2006*; *Kitajima et al., 2006*; *Riedel et al., 2006*; *Tang et al., 2006*; *Liu et al., 2010*; *Foley et al., 2011*). Localization of both PP1 and PP2A at kinetochores depends on microtubule attachment and tension, and changes in these local phosphatase activities may modulate the location of the bistable region of Aurora B activity or exert direct effects on substrates located in the immediate vicinity. These additional mechanisms are not mutually exclusive, and future experiments,

building on our developed in vitro system and quantitative model, should examine how these mechanisms contribute to the establishment and maintenance of gradients at the appropriate spatial scales.

## Materials and methods

### Experimental procedures in vivo

#### Cell culture

HeLa cells were purchased from ATCC and identity was not further authenticated. Cells were cultured at 37°C and 5% $CO_2$ in growth medium: DME (Mediatech, Manassas, VA) with 10% FBS (Atlanta Biologicals, Flowery Branch, GA) and penicillin/streptomycin (Invitrogen, Carlsbad, CA). Cells were shown to be free of *Mycoplasma* contamination by DNA staining. For transient transfections, either Fugene 6 (Promega, Madison, WI) or Lipofectamine 2000 (Invitrogen) were used, following manufacturer's instructions. A HeLa cell line expressing GFP-Aurora B was generated as described previously (*Salimian et al., 2011*), and the expression level of GFP-Aurora B was shown by immunoblotting to be low compared to endogenous Aurora B. Other stable cell lines were generated by recombinase-mediated cassette exchange (RMCE) using the HILO RMCE system (*Khandelia et al., 2011*) as previously described (*Ballister et al., 2014*). For live imaging, cells were grown on 22 x 22 mm glass coverslips (no. 1.5; Thermo Fisher Scientific, Waltham, MA) coated with poly-D-lysine (Sigma-Aldrich, Allentown, PA), and coverslips were mounted in magnetic chambers (CM-S22-1, LCI, Chamlide, Seoul, South Korea) for imaging. Alternatively, cells were grown on poly-D-lysine coated coverslip bottom dishes (MatTek, Ashland, MA). Before imaging, cells were transferred to L-15 medium without Phenol Red (Invitrogen) supplemented with 10% FBS and penicillin/streptomycin. Inhibitors were used at the following concentrations: 100 ng/mL nocodazole, 10 μM MG132 (proteasome inhibitor), 175 nm reversine (Mps1 inhibitor), 10 μM S-trityl-L-cysteine (STLC) or 100 μM monastrol (kinesin-5 inhibitors). The concentration of the Aurora B inhibitor ZM447439 was varied as described in the text.

#### Live cell imaging and analysis

For live imaging of anaphase, a stable cell line was generated expressing the chromatin-targeted Aurora B FRET sensor (*Fuller et al., 2008*), and the cells were transiently transfected with Aurora B-mCherry (mCherry at the C-terminus of human Aurora B). Images were acquired with a spinning disk confocal microscope: an inverted microscope (DMI4000; Leica, Wetzlar, Germany) equipped with a 100x 1.4 NA objective, an XY Piezo-Z stage (Applied Scientific Instrumentation, Eugene, OR), a spinning disk (Yokogawa, Tokyo, Japan), an electron multiplier charge-coupled device camera (ImageEM; Hamamatsu Photonics, Middlesex, NJ), and a laser merge module equipped with 444, 488, and 593-nm lasers (LMM5; Spectral Applied Research, Richmond Hill, Canada) controlled by MetaMorph software (Molecular Devices, Sunnyvale, CA). Temperature was maintained at ~35°C using an environmental chamber (Incubator BL; PeCon, Germany). For FRET imaging, CFP was excited at 444 nm, and CFP and YFP emissions were acquired simultaneously with a beam splitter (Dual-View; Optical Insights, Tucson, AZ). Anaphase was induced by addition of reversine, and images were acquired at 80 sec intervals, 5 z-slices with 1 μm spacing at each time point.

For rapamycin-induced recruitment of Aurora B to centromeres, a stable cell line (pERB261) was created with the following components: (1) the DNA binding domain of CENP-B (CENP-B[DBD]) fused to a tandem trimer of FKBP, constitutively expressed; (2) a miRNA targeting the 3' UTR of endogenous FKBP (*Ballister et al., 2014*), constitutively expressed; (3) a miRNA-based shRNA targeting endogenous INCENP, inducibly expressed; (4) mCherry-INbox-FRB: mCherry fused to INbox, a C-terminal fragment of INCENP (amino acids 818–918 of human INCENP) that binds and activates Aurora B (*Adams et al., 2000*; *Bishop and Schumacher, 2002*; *Bolton et al., 2002*; *Sessa et al., 2005*), and to FRB, inducibly expressed. The following oligos were used for the miRNA targeting the sequence CAGAGGAACCAGATGCTCAT in the endogenous INCENP transcript: 5'-TGCTGA TGAGCATCTGGTTCCTCTGCGTTTTGGCCACTG-ACTGACGCAGAGGACAGATGCTCAT-3' and 5'-CCTGATGAGCATCTGTCCTCTGCGTCAGTCAG-TGGCCAAAACGCAGAGGAACCAGATGCTCA TC-3'. FKBP and FRB are dimerization domains that bind rapamycin, and endogenous FKBP depletion improves rapamycin dimerization efficiency (*Ballister et al., 2014*). 125 ng/mL doxycycline was

added to the growth medium 2 days prior to experiment to induce expression of mCherry-INbox-FRB and the miRNA against endogenous INCENP. INCENP depletion was confirmed by immunofluorescence (data not shown). For live imaging, cells were treated with the kinesin-5 inhibitor S-trityl-L-cysteine (STLC) at least 2 hr before imaging, or with nocodazole at least 1 hr before imaging. Cells were imaged using the spinning disk confocal microscope described above. Rapamycin was added on the microscope by medium exchange to induce Aurora B recruitment to centromeres. Images were acquired at 5 min intervals, 3 z-slices with 0.5 µm spacing at each time point. Cells that were not treated doxycycline, in which INCENP is not depleted, were used to measure maximal phosphorylation.

For bistability experiments in vivo, cells were transiently transfected with the Aurora B FRET sensor targeted either to chromatin (*Fuller et al., 2008*) or to centromeres (*Liu et al., 2009*). In the latter case, mTFP1 is used as the FRET donor rather than CFP. For the centromere-targeted sensor, cells were incubated with MG132 for 30 min, then ZM447439 was added at the indicated concentrations and cells incubated for a further 1 hr. Cells were transferred to L-15 + MG132 +/- ZM447439, and images of ~25 cells were acquired for each condition within 25 min of mounting the coverslip on the microscope. For the control case (no ZM447439), cells were imaged after the initial 30 min MG132 incubation. CFP and YFP images for FRET were acquired with a spinning disk confocal microscope as described above. Five z-slices were acquired for each cell with 0.5 µm spacing. For the chromatin-targeted sensor, cells were incubated with the indicated concentration of ZM447439, together with either nocodazole or MG132 and monastrol, for 1 hr before imaging. Images were acquired using a 100x 1.4 NA objective on an inverted widefield fluorescence microscope (DMI6000, Leica Microsystems) equipped with an automated XYZ stage (Ludl, Hawthorne, NY), an electron multiplier charge-coupled device camera (QuantEM, 512 SC; Photometrics, Tucson, AZ), and a SPECTRA X light engine (Lumencor, Beaverton, OR), controlled by Metamorph Software (Molecular Devices), and a stage top incubator (ZILCS; Tokai Hit, Tokyo, Japan) heated at ~35°C. For FRET imaging, CFP (438/32 nm) and YFP (542/27) emissions were acquired sequentially using CFP excitation (438/24 nm).

For hysteresis experiments in vivo, cells were transiently transfected with the chromatin-targeted Aurora B FRET sensor (*Fuller et al., 2008*). Cells were treated with nocodazole and either 0 µM or 1.5 µM ZM447439 for 100 min and imaged live on the widefield microscope described above. After the first set of images was acquired, the concentration of ZM447439 was either increased (from 0 to 0.6 or 1.5 µM) or decreased (from 1.5 to 0 or 0.6 µM).

For analysis of FRET images, the YFP/CFP emission ratio was calculated using a custom MatLab script, and projection images were prepared as described (*Fuller et al., 2008*). Images represent the mean FRET ratio calculated over a z-stack. For the rapamycin-induced recruitment experiments, cells with FRET ratio less than 1.7 at the first time point, likely due to poor knockdown of endogenous INCENP in a minority of cells, were excluded from the analysis. For the bistability experiments, FRET measurements under three different experimental conditions (see text for details) produced bimodal distributions, but the relative positions of the peaks with 'low' and 'high' FRET ratios were different in different experiments due to different probe locations (H2B or CENP-B) and imaging conditions. To plot these FRET values together (*Figure 5B*), the data points for each experimental condition were split into two groups ('low' and 'high'). The mean FRET values were calculated for each group ($M_{low}$ and $M_{high}$) and the absolute FRET values ($F_{abs}$) for each cell and experimental condition were then normalized using the following expression: $(F_{abs} - M_{low})/(M_{high} - M_{low})$. To plot the normalized sensor phosphorylation values in *Figures 1A*, *5D* and *7B*, inverse normalization of FRET signal was performed using the following expression: $(m_{max} - F_{abs})/(m_{max} - m_{min})$, where $m_{max}$ and $m_{min}$ are the maximum and minimum FRET values in this dataset. The normalized values for each concentration of ZM447439 were averaged before plotting.

For analysis of wave propagation of FRET sensor phosphorylation, cells expressing the chromatin-targeted sensor were arrested with monastrol (*Figure 7C*). A line on each visible chromosome was drawn starting from the centromere, extending towards the periphery along the chromosome arm. The time when each position along this line reached 50% FRET ratio between the maximal FRET ratio (before washout) and minimal FRET ratio (after sensor phosphorylation stopped changing) was recorded. The time point when the FRET ratio reached 50% at the centromere (x = 0) was set as t = 0.

### Fixed cell analysis

To measure Aurora B and PP1γ localization and phospho-INCENP at different concentrations of ZM447439, cells were incubated for two hours with nocodazole with the indicated ZM447439 concentration, then fixed for 10 min with 4% formaldehyde (Amresco, Solon, OH) in DPBS (Corning, Corning, NY). The following antibodies were used: 1:100 mouse anti-Aurora B (AIM-1, BD Biosciences, Franklin Lakes, NJ), 1:1000 rabbit anti-phospho-INCENP (*Salimian et al., 2011*), and 1:200 Alexa Fluor 488 and Alexa Fluor 594 secondary antibodies (Invitrogen). For PP1 localization, a cell line expressing PP1γ-GFP was used (*Liu et al., 2010*). Images were acquired on the spinning disk confocal described above.

## Experimental procedures in vitro

### Aurora B kinase purification

A bicistronic construct containing the DNA sequence of Aurora B$^{60-361}$ and INCENP$^{790-856}$ (*Figure 2—figure supplement 1A*) from *Xenopus laevis* was amplified by PCR using the following primers: 5'-GGGCCCGGATCCTCCTCCAGCGTTCCAGG-3' and 5'-CCCGGGGCGGCCGC-TTAAGGGGAGTGCCATACAGC-3'. The template for the PCR reaction was a PGEX-6P plasmid containing a bicistronic message of full length Aurora B and INCENP$^{790-856}$ (a gift of Dr. Stukenberg). The resulting PCR product was cloned into the *Bam*HI/*Not*I sites of a pRSF duet vector customized to have a GST tag in frame with Aurora B, separated by a TEV cutting site. The Aurora B:INCENP$^{790-856}$ complex was expressed in *E. coli* strain BL21 (pLys) and purified in two steps. First, conventional affinity chromatography was used with glutathione beads, and the protein complex was eluted by cutting with TEV protease (as in *Sessa et al., 2005*). Second, size exclusion chromatography with a Superdex S-200 column was used. Aurora B:INCENP$^{790-856}$ eluted in a single peak from the size exclusion column (*Figure 2—figure supplement 1B*); the eluted protein complex was concentrated, supplemented with 50% glycerol and stored at -20°C after diluting to final protein concentration 50 μM (in Tris-HCl (pH 7.5) 12 mM, NaCl 150 mM, DTT 2 mM and 50% glycerol). The Aurora B:INCENP$^{790-856}$ protein complex was used for all in vitro studies and for simplicity we refer to it as 'Aurora B kinase'.

### Measurement of kinase activity

The Aurora B kinase activity was measured using a commercial chemosensor (Omnia KNZ1161, Thermo Fisher Scientific). This sensor contains a peptide with the consensus recognition sequence for Aurora kinases (RRF-S-L) conjugated with a Sox fluorescent probe. Upon phosphorylation of serine residue, the Sox probe binds soluble magnesium and experiences chelation-enhanced fluorescence. Emission at 485 nm was monitored (*Figure 2—figure supplement 2B*) with a Fluoromax 3 spectro-fluorimeter (Jobin Yvon, Horiba, Edison, NJ) or with a 814 Photomultiplier Detection System (Photon Technology International, Edison, NJ) using a quartz fluorimeter cuvette (105.251-QS, Hellma Analytics, Müllheim, Germany); the excitation wavelength was 400 nm. In routine experiments, the fluorescence of 10 μM chemosensor was measured for at least 10 min in 50 μl kinase assay buffer: Tris-HCL (pH 7.5) 50 mM, NaCl 50 mM, MgCl$_2$ 12 mM, ATP 4 mM, phosphonoacetic acid 10 mM (Sigma-Aldrich, cat#284270), DTT 2 mM, Brij23 0.01%, EGTA 0.5 mM, BSA 0.5 mg/ml. Aurora B kinase was added at 10 nM final concentration (unless stated otherwise) and the cuvette was sealed to avoid evaporation. Final concentration of glycerol did not exceed 1%. To determine fluorescence intensity of the phosphorylated chemosensor (product), the phosphorylation reaction was observed to reach a plateau, such that the product concentration was assumed to equal the initial concentration of the unphosphorylated chemosensor. Standard curves for substrate and product were used to convert fluorescence intensity counts per second (cps) into peptide concentrations; the slopes of these curves were 6520 ± 80 and 15,000 ± 400 cps/μM, respectively (*Figure 2—figure supplement 2C*). The phosphorylation curves were analyzed to extract the initial rates of phosphorylation; these rates were plotted as a function of substrate concentration and fitted to the Michaelis-Menten equation with Prism (GraphPad, La Jolla, CA) software (*Figure 2—figure supplement 2D*). The Lineweaver–Burk plot (*Figure 2—figure supplement 2E*) and Hanes–Woolf plot (*Figure 2—figure supplement 2F*) for these data show good linearity, confirming Michaelis–Menten mechanism for chemosensor phosphorylation (*Bisswanger, 2008*). The value of $V_{max}$ was determined from the Michaelis–Menten curve in *Figure 2—figure supplement 2D*; the value of $k_{cat}$ was

calculated from $V_{\mathrm{max}}$ for 10 nM of Aurora B (**Table 2**), as used in our experiments. The slope of the linear fit for the Lineweaver–Burk plot was used to calculate $\frac{k_{cat}}{K_M}$. The initial rate for 10 µM chemosensor phosphorylation by active Aurora B (5 nM) was 140 times faster than by the partially active Aurora B (5 nM); number of independent experiments $\geq$ 3.

## Characterization of Aurora B autoactivation

To evaluate kinetics of Aurora B autoactivation, purified Aurora B was first treated with λ protein phosphatase ($8 \cdot 10^8$ units/g; P0753S, New England BioLabs, Ipswich, MA) to obtain the 'partially active' Aurora B. Activity of the phosphatase (units) was converted to concentration (µM) using the conversion factor $5 \cdot 10^{-14}$ mol/units. Aurora B (16 µM) was incubated for 2 hr at 30°C with 0.2 µM phosphatase in the 'inactivation' buffer, which was same as 'kinase assay' buffer but with no ATP, phosphonoacetic acid, EGTA, BSA, and supplemented with $MnCl_2$ 100 µM. Since the partially active kinase was highly inefficient in phosphorylating the chemosensor substrate, at low kinase concentration the changes in kinase activity due to autophosphorylation were examined directly in the presence of chemosensor. Partially active Aurora B kinase was diluted to 0.16, 0.5 or 1.5 µM in the 'kinase assay' buffer without BSA and EGTA ('activation' buffer), which caused full phosphatase inhibition (**Figure 2—figure supplement 1C**). The reaction mixture was then supplemented with Omnia chemosensor 20 µM, and phosphorylation kinetics were measured (**Figure 2A**).

At high Aurora B kinase concentration the chemosensor substrate becomes rapidly depleted, so we performed activation reaction separately from the chemosensor phosphorylation. The partially active Aurora B (4 µM) was incubated in 'activation' buffer to allow autoactivation to take place (**Figure 2D**). As a control, the fully active kinase was first preincubated in the 'inactivation' buffer with no phosphatase, then incubated in the 'activation' buffer, as done with the partially active Aurora B. Both, experimental and control samples were subjected to dialysis in 3,500 MWCO mini dialysis units (Slide-A-Lyzer, Thermo Fischer Scientific) at 30°C in 100 ml of activation buffer with glycerol 3% and $MnCl_2$ 25 µM. Dialysis was required to avoid ATP depletion during Aurora B autoactivation. The degree of Aurora B activation at different times was determined by taking an aliquot of the autoactivation reaction diluted in 'kinase assay' buffer to obtain 10–80 nM of final Aurora B concentration. This dilution essentially stopped all autoactivation reactions, so the initial rate of chemosensor phosphorylation (< 10% phosphorylation), could be determined. The initial rate in samples with partially active kinase was compared with the initial rate in control samples taken at the same time, and the concentration of active kinase was determined (**Figure 2D**).

## Hysteresis experiments

In one set of experiments, purified Aurora B, which is fully active, was used ('initially high' experiment); in the second set, the partially active Aurora B was used ('initially low' experiment). For the 'initially high' experiment, 8 µM Aurora B was mixed with the phosphatase (0.25–0.8 µM range) in 'hysteresis' buffer: Tris-HC (pH 7.5) 25 mM, NaCl 100 mM, $MgCl_2$ 5 mM, ATP 4 mM, DTT 2 mM, $MnCl_2$ 100 µM, Brij23 0.01%. Immediately after mixing, 12 µl of the reaction mixture were placed in mini dialysis units in 100 ml of 'hysteresis' buffer, final glycerol concentration from adding enzyme stock was 9%. At different times, small aliquots of the reaction mix were taken and diluted to the final kinase concentration of 5 nM. The kinase assay was carried out, as described above, using 10 µM of custom made chemosensor, which contained the same peptide as the commercial sensor but the Sox fluorescent probe was conjugated via cysteine bond (**González-Vera et al., 2009**). Aurora B kinase phosphorylates this chemosensor faster than the commercial one, with $K_M$ = 55 µM and $k_{cat}$ = 19 s$^{-1}$. The initial rate values in **Figure 4C** were determined after the Aurora B-phosphatase coupled system reached steady state ($\geq$60 min). Prior to plotting, these steady-state rates were normalized to the initial rate measured with the fully active kinase in the absence of phosphatase. For the 'initially low' experiments, purified Aurora B was pretreated with phosphatase (0.15–0.5 µM range) for 30 min in 'hysteresis' buffer lacking ATP. Then, ATP at 4 mM was added and the reaction mixture was incubated in mini dialysis units. The initial rates for Aurora B kinase reactions were determined and plotted as in 'initially high' experiments.

## Theoretical modeling of experiments in vitro

### General model description

All biochemical reactions were described using equations of enzyme kinetics with reactants, enzymes and products denoted using symbols in *Table 1* and rate constants in *Table 2*. We modeled a stable Aurora B complex consisting of Aurora B kinase and its regulatory subunit INCENP. This complex can be phosphorylated at several sites (*Sessa et al., 2005*), but all modeling graphs in this paper show results obtained with a simplified model that incorporates a single phosphorylation site. Below, we also provide analysis of the model with 2 phosphosites and show that the main conclusions from our in vitro experiments do not change. The single site model assumes that:

1. Aurora B kinase has two states: the non-phosphorylated state (A) that is partially active, and the phosphorylated state (A*) with maximal activity.
2. Partially active Aurora B kinase can activate itself in cis. This assumption is justified by our findings in *Figure 2C*. We also assume for simplicity that this reaction is single-step and not reversible.
3. Phosphorylated active Aurora B kinase can phosphorylate the partially active Aurora B with Michaelis-Menten kinetics.
4. Phosphorylated active Aurora B kinase can phosphorylate its substrate (e.g. unphosphorylated chemosensor) with Michaelis-Menten kinetics.
5. Partially active Aurora B has no activity towards the substrate. This assumption is justified by our finding that partially active Aurora B has > 100-times lower activity than the active Aurora B.

Following system of equations described Aurora B activation in the presence of substrate:

$$
\begin{cases}
A \xrightarrow{k_{cis}} A^* \\
A + A^* \xleftrightarrow{k_f^a,\ k_r^a} [AA^*] \xrightarrow{k_{cat}^a} A^* + A^* \\
S + A^* \xleftrightarrow{k_f,\ k_r} [SA^*] \xrightarrow{k_{cat}} P + A^*
\end{cases}
\tag{1}
$$

This reaction scheme is fully described by the following system of ordinary differential equations (ODEs):

$$
\begin{cases}
dA^*/dt = A \cdot k_{cis} + [AA^*] \cdot (2k_{cat}^a + k_r^a) - A^* \cdot A \cdot k_f^a + [SA^*] \cdot (k_{cat} + k_r) - S \cdot A^* \cdot k_f \\
d[AA^*]/dt = A^* \cdot A \cdot k_f^a - [AA^*] \cdot (k_{cat}^a + k_r^a) \\
d[SA^*]/dt = S \cdot A^* \cdot k_f - [SA^*] \cdot (k_r + k_{cat}) \\
dS/dt = -S \cdot A^* \cdot k_f + [SA^*] \cdot k_r \\
dP/dt = [SA^*] \cdot k_{cat}
\end{cases}
\tag{2}
$$

See *Table 1* and *2* for details; subscripts *f* and *r* correspond to forward and reverse reactions, respectively; $k_r^a$ is the rate constant for the reverse reaction of in trans autoactivation; $k_r$ is the rate constant for the dissociation of the kinase-substrate complex.

**Table 1.** Glossary of symbols used in this work. Unless stated otherwise, symbols refer to concentrations of enzymes or enzyme-substrate complexes (µM).

| Symbol | Description |
|---|---|
| A | partially active Aurora B kinase |
| A* | active Aurora B kinase |
| PPase | phosphatase |
| [A*PPase] | enzyme-substrate complex for the phosphatase and active Aurora B kinase |
| [AA*] | enzyme-substrate complex for the active and partially active Aurora B kinase molecules |
| S | chemosensor (substrate) |
| P | phosphorylated chemosensor (product) |
| [SA*] | enzyme-substrate complex for the active Aurora B kinase and chemosensor |

**Table 2.** Enzyme kinetic constants used in this work. Values in brackets correspond to measurements with the custom made chemosensor.

| Symbol | Description | Value | Units | Source |
|---|---|---|---|---|
| $k_{cat}$ | catalytic rate constant for active Aurora B kinase towards chemosensor | 19 (19) | s$^{-1}$ | *Figure 2—figure supplement 2D* |
| $\frac{k_{cat}}{K_M}$ | catalytic efficiency of active Aurora B towards chemosensor | $6 \times 10^{-2}$ ($3 \times 10^{-1}$) | s$^{-1}$µM$^{-1}$ | *Figure 2—figure supplement 2E* |
| $K_M$ | Michaelis constant of active Aurora B kinase towards chemosensor | 320 (55) | µM | this work |
| $k_f$ | rate constant for the formation of the enzyme-substrate complex of active Aurora B kinase and chemosensor | 50 | µM$^{-1}$ s$^{-1}$ | estimated based on *Wassaf et al., 2006*; *Schreiber et al., 2009* |
| $k_f^a$ | rate constant for the formation of the enzyme-substrate complex of active and partially active Aurora B kinase molecules | 0.1 | µM$^{-1}$ s$^{-1}$ | estimated based on *Schlosshauer and Baker, 2004*; *Schreiber et al., 2009* |
| $K_M^a$ | Michaelis constant of active Aurora B kinase towards partially active Aurora B kinase | 51 | µM | fitting (*Figure 2A,D*) |
| $k_{cat}^a$ | catalytic rate constant for active Aurora B kinase towards the partially active Aurora B kinase | $2.7 \times 10^{-2}$ | s$^{-1}$ | fitting (*Figure 2A,D*) |
| $k_{cis}$ | rate constant for Aurora B kinase activation by *cis* mechanism | $7.29 \times 10^{-6}$ | s$^{-1}$ | fitting (*Figure 2A,D*) |
| $k_f^p$ | rate constant for the formation of the enzyme-substrate complex of λ protein phosphatase and active Aurora B kinase | 0.6 | µM$^{-1}$ s$^{-1}$ | estimated based on *Schlosshauer and Baker, 2004*; *Schreiber et al., 2009* |
| $K_M^p$ | Michaelis constant of the λ protein phosphatase towards active Aurora B kinase | 1.95 | µM | *Figure 4C* |
| $k_{cat}^p$ | catalytic rate constant for λ protein phosphatase towards active Aurora B kinase | $2.4 \times 10^{-2}$ | s$^{-1}$ | *Figure 4C* |

## Determination of parameters of Aurora B autoactivation

Several enzymatic constants for system (2) could be determined from the literature. We used $k_f$ = 50 µM$^{-1}$ s$^{-1}$ based on the known correlation between the rate of enzyme-substrate complex formation and complex size (*Wassaf et al., 2006*; *Schreiber et al., 2009*). The value of $k_r = (k_f K_M) - k_{cat}$. For the $k_f^a$ rate constant of enzymatic complex formation for the in trans reaction, we used the value reported for the enzyme-substrate pair with the closest radius of gyration to Aurora B, which we estimated to be 1.97 nm, based on the PDB 2BFY structure (*Schlosshauer and Baker, 2004*; *Sessa et al. 2005*; http://www.scfbio-iitd.res.in/software/proteomics/rg.jsp). Based on these calculations, $k_f^a$ = 0.1 µM$^{-1}$ s$^{-1}$. We determined constants for Aurora B phosphorylation of chemosensor substrate as described in the section 'Measurement of kinase activity'.

To determine the $K_M^a$, $k_{cat}^a$ and $k_{cis}$ rate constants for Aurora B autoactivation we fitted experimental data in *Figure 2A and D* by numerically solving equation system (2) with custom-made software 'Parameter Estimation and Fitting Tool' (PEFT, described below), leading to solid lines in *Figure 2A and D*. The value of $k_r^a$ was determined analogously to $k_r$.

## Theoretical analysis of bistability and hysteresis in a coupled kinase-phosphatase system

To describe the coupled Aurora B kinase-phosphatase system, we assume that active Aurora B kinase is a substrate for phosphatase and is dephosphorylated with Michaelis-Menten kinetics. We

also assume that Aurora B kinase in the enzyme-substrate complex cannot be dephosphorylated. The following reaction scheme was used:

$$\begin{cases} A \xrightarrow{k_{cis}} A^* \\ A + A^* \xrightleftharpoons[k_r^a]{k_f^a,\ k_r^a} [AA^*] \xrightarrow{k_{cat}^a} A^* + A^* \\ A^* + PPase \xrightleftharpoons[k_r^p]{k_f^p,\ k_r^p} [A^*PPase] \xrightarrow{k_{cat}^p} A + PPase \end{cases} \tag{3}$$

This reaction scheme led to the following ODEs system:

$$\begin{cases} dA^*/dt = A \cdot k_{cis} + [AA^*] \cdot (2k_{cat}^a + k_r^a) + [A^*PPase] \cdot k_r^p - A^* \cdot A \cdot k_f^a - A^* \ \cdot PPase \ k_f^p \\ d[AA^*]/dt = A \cdot A^* \cdot k_f^a - [AA^*] \cdot (k_{cat}^a + k_r^a) \\ d[A^*PPase]/dt = PPase \cdot A^* \cdot k_f^p - [A^*PPase] \cdot (k_r^p + k_{cat}^p) \end{cases} \tag{4}$$

See **Table 1** and **2** for details; $k_r^p$ is the rate constant for the dissociation of the phosphatase-kinase complex.

First, we studied the steady-state solutions for A* by solving system (4) numerically with Mathematica software (Wolfram Research, Champaign, IL). **Figure 3B** shows the steady-state concentration of active Aurora B as a function of phosphatase concentration for several total Aurora B concentrations. When phosphatase concentration is low, nearly all kinase is active, as expected for autoactivation. With increasing phosphatase concentration, the steady-state concentration of active Aurora B decreases. When total concentration of Aurora B is low this decrease is monotonic. However, for total concentration of Aurora B>3.9 µM, 3 steady state solutions can be seen, demonstrating bistability. The region of parameters for which three steady states coexist (region of bistability) is shown in **Figure 3C**.

The presence of the unstable steady state in this system suggests that its kinetic behavior should depend on a threshold. Kinetic behavior of the system of equations 4 was analyzed using different Aurora B kinase and phosphatase concentrations and ODEs solver of PEFT software. In the region of bistability, the steady-state concentration of active Aurora B depends on the relationship between initial conditions and threshold value. Presence of the threshold is evident from the behavior of this system when initial conditions are chosen close to threshold (**Figure 3D**) and in response to perturbations of the steady state (**Figure 3E**). Calculations for **Figure 3E** started from 8 µM of partially active Aurora B and the systems achieved the steady state with low Aurora B activity. We then simulated additions of small amounts of active Aurora B of different concentration (indicated with arrows in this graph). The first two additions of active Aurora B (0.3 and 0.4 µM) did not increase the fraction of active Aurora B, as the injected active kinase was rapidly inactivated. The last addition of 0.5 µM active Aurora B pushed the system above the threshold, and it rapidly transitioned into the steady state with high activity (**Figure 3E**). To examine theoretically whether this system exhibits hysteresis, solutions for system (4) were found for various levels of phosphatase. Calculation with active Aurora B started from 0 µM phosphatase. Phosphatase concentration was then gradually increased up to 1 µM, then decreased down to 0 µM with the same speed (shown with arrows in **Figure 3F**). The slow speed of these transitions ensured that the system reached quasi steady state at all phosphatase concentrations. The total calculation time for one such cycle equaled 100 hr of the reaction time. The resulting graph in **Figure 3F** shows that trajectories for the motion toward high and low phosphatase concentrations superimposed completely in the areas where the system has a single steady state. In the bistable region, the trajectories bifurcate: when phosphatase concentration is increasing, the system follows the upper branch, but it follows the lower branch when phosphatase is reduced, giving rise to a hysteresis loop.

## Analysis of the bistability dependence on phosphatase catalytic constants

Since kinetic properties of the phosphatase(s) that inactivate Aurora B kinase in cells are not yet known, we investigated how model behavior depends on catalytic constants for phosphatase, $K_M^p$ and $k_{cat}^p$. Calculations with varying phosphatase concentration were carried out with ODEs solver using 'initial high' and 'initial low' starting conditions, and hysteresis loops were observed for all tested values of $K_M^p$ (**Figure 3—figure supplement 1B**). The bistability regions were present in a similar range of phosphatase concentrations, but the region became slightly more narrow with increasing $K_M^p$. Changes in catalytic rate $k_{cat}^p$, as expected, did not change the overall shape of these

curves but shifted them to a different range of phosphatase concentration. When these curves are plotted versus the phosphatase concentration divided by the corresponding $k_{cat}^p$, they overlap completely (*Figure 3—figure supplement 1C*). These data indicate that difference in catalytic constants for phosphatase can be compensated by adjusting phosphatase concentration, such that the major model predictions remain unchanged. We conclude that bistability of coupled Aurora B kinase-phosphatase system can be observed for various phosphatases, and the presence of non-linear regimes depends mostly on the biochemical properties of Aurora B kinase.

## Modeling of experiments with bistability and hysteresis in vitro

System (4) was solved with PEFT assuming the initial condition with 100% active Aurora B concentration to simulate the 'initial high' experimental scheme, and that 100% of total Aurora B kinase was partially active to simulate the 'initial low' experiment. All parameters that describe Aurora B autoactivation were fixed as in (*Table 2*). Initially, concentrations of all enzyme-substrate complexes were zero, and total Aurora B concentration was 8 µM. ODEs solver was applied independently for each phosphatase concentration in the range of 0.01 µM to 0.8 µM with 0.01 µM step. Total simulation time was 1.5 hr per each phosphatase concentration and initial condition. To obtain the fraction of active Aurora B kinase we computed the sum of concentrations of active Aurora B kinase in a free form and as a part of the enzymatic complex with partially active Aurora B, then divided this sum by the total Aurora B concentration. For these calculations we used the value for $k_f^p$ catalytic rate constant of enzymatic complex formation 0.6 µM⁻¹ s⁻¹, which we calculated analogously to $k_f^a$ constant for Aurora B (see section 'Determination of parameters of Aurora B autoactivation'). The radius of gyration for λ phosphatase was 1.7 nm (PDB structure 1G5B; *Voegtli et al., 2000*). The values of $K_M^p$ and $k_{cat}^p$ were chosen to match experimental data in *Figure 4C*. The value of $k_r^p$ was determined analogously to $k_r$.

## Model of Aurora B autoactivation via two phosphorylation sites

Work by (*Sessa et al., 2005*) suggests that at least two phosphorylation sites are involved in Aurora B kinase autoactivation: one located in the Aurora B kinase autoactivation loop and the second site in the INCENP INbox domain. We constructed a model in which full Aurora B activation requires phosphorylation at two sites based on the following assumptions:

1. The phosphorylation is sequential, i.e. the second site (e.g. in INbox domain) becomes phosphorylated only after the first site has been phosphorylated.
2. Aurora B kinase has three activity levels corresponding to the (i) fully unphosphorylated state $A$, which is partially active, (ii) state $A^\#$ with intermediate activity, which requires phosphorylation at the first site, and (iii) state $A^{\#\#}$ with maximal activity and both sites phosphorylated.
3. Kinase $A$ can activate itself via in cis reaction to convert into the intermediate form $A^\#$ but does not phosphorylate kinase of any other state in trans.
4. Kinase $A$ can be phosphorylated by the $A^\#$ and $A^{\#\#}$ kinases via Michaelis-Menten reactions.
5. Kinase $A^\#$ can be phosphorylated by the $A^\#$ and $A^{\#\#}$ kinases via Michaelis-Menten reactions. The catalytic rate constant of the $A^{\#\#}$ kinase is 7-fold greater than of $A^\#$ kinase, but their Michaelis constants are the same (based on *Sessa et al., 2005*).

Substrate (chemosensor) phosphorylation was described as in the model with one phosphosite (2) with constants listed in *Table 2*, leading to the system of ODEs (5). This model was solved numerically with ODEs solver. Predicted chemosensor product concentration $P$ as a function of time is shown in *Figure 2—figure supplement 3A*. As expected, the two sites model provides a good description to these experimental data with unconstrained best fit constants (*Table 3*). To describe autoactivation of 4 µM Aurora B without chemosensor, we put $S = 0$ and calculated concentration of active Aurora B as a sum of concentrations of $A^\#$ and $A^{\#\#}$ forms (*Figure 2—figure supplement 3E*). To model hysteresis experiments in vitro we supplemented the two sites model (5) with phosphatase reactions. Here, we assumed that kinase forms $A^\#$ and $A^{\#\#}$ could be dephosphorylated by a phosphatase with Michaelis-Menten like kinetics and that these reactions took place in the reverse order relative to the reactions of phosphorylation: $A^{\#\#}$ could only be converted to $A^\#$, which was then converted to $A$:

$A^\#+PPase \rightleftharpoons [A^\#PPase] \rightarrow A+ PPase$ with reaction rate constants $l_f^{A\#PPase}$; $l_r^{A\#PPase}$; $l_{cat}^{A\#PPase}$

$A^{\#\#}+PPase \rightleftarrows [A^{\#\#}PPase] \rightarrow A^{\#} + PPase$ with reaction rate constants $l_f^{A\#\#PPase}$; $l_r^{A\#\#PPase}$; $l_{cat}^{A\#\#PPase}$.

For these calculations, the fitted values of all parameters listed in *Table 3* were used with $l_f^{A\#\#PPase} = l_f^{A\#PPase}$ = 0.6 µM$^{-1}$ s$^{-1}$; $l_r^{A\#PPase} = l_r^{A\#\#PPase}$ = 5.9 s$^{-1}$; $l_{cat}^{A\#PPase} = l_{cat}^{A\#\#PPase}$ = 0.12 s$^{-1}$, also leading to a good match with experimental data (*Figure 2—figure supplement 3F*).

$$
\begin{cases}
\dfrac{dA^{\#}}{dt} &= Al_{cis} + 2[AA^{\#}]l_{cat}^{AA\#} - AA^{\#}l_f^{AA\#} + [AA^{\#}]l_r^{AA\#} + [AA^{\#\#}]l_{cat}^{AA\#\#} + [A^{\#}A^{\#}]l_{cat}^{A\#A\#} - \\[4pt]
&\quad -2(A^{\#}A^{\#}l_f^{A\#A\#} - [A^{\#}A^{\#}]l_r^{A\#A\#}) - A^{\#}A^{\#}l_f^{A\#A\#\#} + \\
&\quad +[A^{\#}A^{\#\#}]l_r^{A\#A\#\#} + [SA^{\#}]l_{cat}^{SA\#} + [SA^{\#}]l_r^{SA\#} - A^{\#}l_f^{SA\#}S \\[8pt]
\dfrac{dA^{\#\#}}{dt} &= [AA^{\#\#}]l_{cat}^{AA\#\#} - AA^{\#\#}l_f^{AA\#\#} + [AA^{\#\#}]l_r^{AA\#\#} + [A^{\#}A^{\#}]l_{cat}^{A\#A\#} + 2[A^{\#}A^{\#\#}]l_{cat}^{A\#A\#\#} - \\[4pt]
&\quad -A^{\#}A^{\#\#}l_f^{A\#A\#\#} + [A^{\#}A^{\#\#}]l_r^{A\#A\#\#} + [SA^{\#\#}]l_{cat}^{SA\#\#} \\
&\quad +[SA^{\#\#}]l_r^{SA\#\#} - A^{\#\#}l_f^{SA\#\#}S \\[8pt]
\dfrac{dA^{\#}A}{dt} &= -[AA^{\#\#}]l_{cat}^{AA\#\#} + AA^{\#\#}l_f^{AA\#\#} - [AA^{\#\#}]l_r^{AA\#\#} \\[8pt]
\dfrac{d[A^{\#}A^{\#}]}{dt} &= -[A^{\#}A^{\#}]l_{cat}^{A\#A\#} + A^{\#}A^{\#}l_f^{A\#A\#} - [A^{\#}A^{\#}]l_r^{A\#A\#} \\[8pt]
\dfrac{dA^{\#\#}A}{dt} &= -[AA^{\#\#}]l_{cat}^{AA\#\#} + AA^{\#\#}l_f^{AA\#\#} - [AA^{\#\#}]l_r^{AA\#\#} \\[8pt]
\dfrac{d[A^{\#}A^{\#\#}]}{dt} &= -[A^{\#}A^{\#\#}]l_{cat}^{A\#A\#\#} + A^{\#}A^{\#\#}l_f^{A\#A\#\#} - [A^{\#}A^{\#\#}]l_r^{A\#A\#\#} \\[8pt]
\dfrac{d[SA^{\#}]}{dt} &= -[SA^{\#}]l_{cat}^{SA\#} - [SA^{\#}]l_r^{SA\#} + A^{\#}l_f^{SA\#}S \\[8pt]
\dfrac{d[SA^{\#\#}]}{dt} &= -[SA^{\#\#}]l_{cat}^{SA\#\#} - [SA^{\#\#}]l_r^{SA\#\#} + A^{\#\#}l_f^{SA\#\#}S \\[8pt]
\dfrac{dS}{dt} &= [SA^{\#}]l_r^{SA\#} + [SA^{\#\#}]l_r^{SA\#\#} - A^{\#}l_f^{SA\#}S - A^{\#\#}l_f^{SA\#\#}S \\[8pt]
\dfrac{dP}{dt} &= [SA^{\#}]l_{cat}^{SA\#} + [SA^{\#\#}]l_{cat}^{SA\#\#}
\end{cases}
\tag{5}
$$

Here, $l_f$ and $l_r$ are the rate constants for enzyme-substrate complex formation and dissociation, respectively; $l_{cat}$ is the catalytic rate constant. Upper indices of the rate constants indicate the corresponding enzyme-substrate complex.

To examine the relationship between the one site and two sites models of Aurora B phosphorylation, we examined changes in concentration of different Aurora B forms during the autoactivation time course. *Figure 2—figure supplement 3C* shows that the intermediate form with one phosphosite $A^{\#}$ appears transiently as an intermediate product, which later becomes converted into the completely phosphorylated form $A^{\#\#}$. We therefore tested if the one site model can be considered as a limiting case of the two sites model. We significantly (500-fold) increased the rate of conversion of $A^{\#}$ into $A^{\#\#}$, thereby reducing the maximal concentration of the intermediate kinase form down to few percent of the total (*Figure 2—figure supplement 3D*). We then identified the set of model parameters (limiting case, *Table 3*) that produced a good match to experiments in *Figure 2—figure supplement 3* panels B and E. Remarkably, the limiting case parameter values were nearly identical to those we obtained with a one site model (*Table 2*). This analysis strongly justifies our use of a simplified model with only one phosphorylation site, as this model can be interpreted as a two site model with a rapidly converting Aurora B form with one phosphorylated site.

## Theoretical modeling of experiments in cells

### Description of the spatial model of Aurora B phosphorylation

The spatial model is based on the homogeneous model of the coupled kinase-phosphatase system described above. Additionally, it incorporates spatial distribution of Aurora B binding sites to mimic

**Table 3.** Aurora B autoactivation model with two phosphorylation sites. Fitted values were obtained using unconstrained fitting, limiting case values correspond to the model in which Aurora B kinase with one phosphosite is converted rapidly into the fully phosphorylated Aurora B form.

| Reaction | Rate constants | Fitted values ($\mu M^{-1}$ $s^{-1}$; $s^{-1}$; $s^{-1}$) | Limiting case values ($\mu M^{-1}$ $s^{-1}$; $s^{-1}$; $s^{-1}$) |
|---|---|---|---|
| $A \rightarrow A^{\#}$ | $l_{cis}$ | $6.0 \times 10^{-5}$ | $7.29 \times 10^{-6}$ |
| $A+A^{\#} \rightleftharpoons [AA^{\#}] \rightarrow A+A^{\#}$ | $l_f^{AA}$; $l_r^{AA}$; $l_{cat}^{AA}$ | 0.1; 10; $1.1 \times 10^{-2}$ | 0.1; 5.1; $3.9 \times 10^{-3}$ |
| $A^{\#}+A^{\#} \rightleftharpoons [A^{\#}A^{\#}] \rightarrow A^{\#\#}+A^{\#}$ | $l_f^{AA}$; $l_r^{AA}$; $l_{cat}^{AA}$ | 0.1; 10; $1.1 \times 10^{-2}$ | 50; 5.1; 1.95 |
| $A+A^{\#\#} \rightleftharpoons [AA^{\#\#}] \rightarrow A^{\#}+A^{\#\#}$ | $l_f^{AA}$; $l_r^{AA}$; $l_{cat}^{AA}$ | 0.1; 9.8; $8.0 \times 10^{-2}$ | 0.1; 5.1; $2.7 \times 10^{-2}$ |
| $A^{\#}+A^{\#\#} \rightleftharpoons [A^{\#}A^{\#\#}] \rightarrow A^{\#\#}+A^{\#\#}$ | $l_f^{AA}$; $l_r^{AA}$; $l_{cat}^{AA}$ | 0.1; 9.8; $8.0 \times 10^{-2}$ | 0.1; 5.1; $2.7 \times 10^{-2}$ |
| $S+A^{\#} \rightleftharpoons [SA^{\#}] \rightarrow P+A^{\#}$ | $l_f^{SA}$; $l_r^{SA}$; $l_{cat}^{SA}$ | 50; $1.6 \times 10^4$; 4.7 | 50; $1.6 \times 10^4$; 4.7 |
| $S+A^{\#\#} \rightleftharpoons [SA^{\#\#}] \rightarrow P+A^{\#\#}$ | $l_f^{SA}$; $l_r^{SA}$; $l_{cat}^{SA}$ | 50; $1.6 \times 10^4$; 19 | 50; $1.6 \times 10^4$; 19 |

Aurora B localization to chromatin in prometaphase and metaphase cells. Aurora B distribution along the centromere-kinetochore axis was derived from the Aurora B fluorescent intensity profile seen at metaphase kinetochores ( Figure 2B in *Liu et al., 2009*). The experimental distribution was deconvolved using a point spread function with a width 0.5 $\lambda$/NA = 210 nm, where $\lambda$ = 594 nm is the excitation wavelength and *NA* = 1.4 is the objective numerical aperture. Normalized fluorescence intensity was converted into concentration using the estimated average concentration of the chromatin-bound Aurora B pool, as described below in section 'Choice of model parameters'. The resulting function *profile1*(x) is shown in *Figure 6—figure supplement 1B*. Aurora B distribution along the chromosome arms, *profile2*(x) (*Figure 6—figure supplement 1C*) was obtained by measuring intensity of chromatin-bound signal in monastrol-arrested HeLa cells expressing GFP-Aurora B (*Figure 6—figure supplement 1* panels D and E).

The following model postulates were used:

1. Both partially active and fully active Aurora B molecules could bind to chromatin binding sites and unbind with kinetic constants $k_{on}$ and $k_{off}$.
2. Soluble Aurora B kinase diffused with diffusion coefficient *D*.
3. Both soluble and chromatin-bound Aurora B kinase could phosphorylate itself in cis and in trans
4. Phosphatase was soluble and could dephosphorylate soluble and bound Aurora B kinase. Catalytic rate constant for this reaction were as in the homogeneous in vitro model (*Table 2*). Phosphatase diffused with diffusion coefficient *D*.

Thus, the spatial model equations incorporate equations for the homogeneous system, reactions for the chromatin-bound Aurora B kinase and the diffusion term for soluble components:

$$
\begin{cases}
\partial A^*/\partial t = A \cdot k_{cis} + [AA^*] \cdot (2k_{cat}^a + k_r^a) - A^* \cdot A \cdot k_f^a - A^* \cdot B \cdot k_f^a + [AB^*] \cdot k_{cat}^a + \\
+ [BA^*] \cdot (k_{cat}^a + k_r^a) + [A^*PPase] \cdot k_r^P - A^* \cdot PPase \cdot k_f^p + B^* \cdot k_{off} - A^* \cdot Sites \cdot k_{on} + D \cdot \partial^2 A^* /\partial x^2 = F_1(x,t) \\
\partial B/\partial t = B \cdot k_{cis} + [BB^*] \cdot (2k_{cat}^a + k_r^b) - B^* \cdot A \cdot k_f^a - B^* \cdot B \cdot k_f^b + [BA^*] \cdot k_{cat}^a + [AB^*] \cdot (k_{cat}^a + k_r^a) + \\
+ [B^*PPase] \cdot k_r^P - B^* \cdot PPase \cdot k_f^p - B^* \cdot k_{off} + A^* \cdot Sites \cdot k_{on} = F_2(x,t) \\
\partial A/\partial t = -A \cdot k_{cis} + ([AA^*] + [AB^*]) \cdot k_r^a - A \cdot (A^* + B^*) \cdot k_f^a + [A^*PPase] \cdot k_{cat}^P + \\
+ B \cdot k_{off} - A \cdot Sites \cdot k_{on} + D \cdot \partial^2 A/\partial x^2 \\
\partial B/\partial t = -B \cdot k_{cis} + [BA^*] \cdot k_r^a + [BB^*] \cdot k_r^b - B \cdot A^* \cdot k_f^a + B \cdot B^* \cdot k_f^b + [B^*PPase] \cdot k_{cat}^P - B \cdot k_{off} + A \cdot Sites \cdot k_{on} \\
\partial [AA^*]/\partial t = A \cdot A^* \cdot k_f^a - [AA^*] \cdot (k_{cat}^a + k_r^a) + D \cdot \partial^2 [AA^*]/\partial x^2 \\
\partial [BA^*]/\partial t = B \cdot A^* \cdot k_f^a - [BA^*] \cdot (k_{cat}^a + k_r^a) \\
\partial [AB^*]/\partial t = A \cdot B^* \cdot k_f^a - [AB^*] \cdot (k_{cat}^a + k_r^a) \\
\partial [BB^*]/\partial t = B \cdot B^* \cdot k_f^b - [BB^*] \cdot (k_{cat}^a + k_r^b) \\
\partial [A^*PPase]/\partial t = PPase \cdot A^* \cdot k_f^p - [A^*PPase] \cdot (k_r^p + k_{cat}^p) + D \cdot \partial^2 [A^*PPase]/\partial x^2 \\
\partial [B^*PPase]/\partial t = PPase \cdot B^* \cdot k_f^p - [B^*PPase] \cdot (k_r^p + k_{cat}^p) \\
\partial Sites/\partial t = (B + B^*) \cdot k_{off} - (A + A^*) \cdot Sites \cdot k_{on}
\end{cases}
$$

$$(6)$$

See *Table 1* and *2* for details; *B\** (*B*) is concentration of bound active (partially active) Aurora B; [*XY*] is enzyme-substrate complex between *X* substrate and *Y* enzyme; '*Sites*' is concentration of free Aurora B binding sites.

For simplicity, simulations were carried out in one dimension and two spatial axes were examined independently: along the chromosome arms axis and along the centromere-kinetochore axis (*Figure 6A*). Boundary conditions were chosen to avoid the flow of soluble components:

$$
\begin{cases}
\partial A^*/\partial x|_{x=-R,R} = 0 \\
\partial A/\partial x|_{x=-R,R} = 0 \\
\partial [AA^*]/\partial x|_{x=-R,R} = 0 \\
\partial [A^*PPase]/\partial x|_{x=-R,R} = 0 \\
\partial PPase/\partial x|_{x=-R,R} = 0
\end{cases}
$$

$$(7)$$

where *x* corresponds to the size of the simulated spatial interval from – *R* to *R*.

System of equations (6) was solved numerically using Mathematica software (Wolfram Research) with boundary conditions (7). Initial conditions were varied for different types of experiments, as described below. For each spatial direction we solved the one dimensional problem using distance from centroid (x) as the one dimensional coordinate and initial conditions for Aurora B binding sites *profile1*(x) or *profile2*(x).

## Choice of model parameters

To quantify the spatial distribution of Aurora B kinase binding sites on chromatin, we used the reported soluble Aurora B concentration in cells $C_{sol}$ = 8.6 nM (*Mahen et al., 2014*) and the estimated volume of 46 centromeres (as in human cells) $V^{tot}_{cent}$ = 46 · 0.5 µm · 0.5 µm · 2 µm = 23 µm$^3$. About 75% of total cellular Aurora B is in the bound form (*Mahen et al., 2014*); therefore 75 · $C_{sol}$ · $V_{cell}$ = 25 · $C_{bound}$ · $V^{tot}_{cent}$, where $C_{bound}$ is average concentration of bound Aurora B, $V_{cell}$ = 5,800 µm$^3$ is cell volume (*Mahen et al., 2014*). Therefore, $C_{bound}$ = 6.4 µM, leading to quantitative profiles in *Figure 6—figure supplement 1* panels B and C with a peak Aurora B concentration 10 µM.

Kinetic parameters of the autoactivation of soluble Aurora B kinase were the same as in the homogeneous model (*Table 2*). Autoactivation parameters for bound Aurora B were the same as for soluble kinase, except the value of $k_f^b$ = 0.01 $k_f^a$ to account for the sterically limited interactions between bound Aurora B kinase molecules. Dissociation constant $k_{off}$ = 0.014 s$^{-1}$ for Aurora B was based on the previously measured CPC turnover time of 50 s (*Murata-Hori and Wang, 2002*). Using $K_D$ = 4.8 nM for the CPC complex (*Mahen et al., 2014*), the association rate $k_{on}$ for Aurora B kinase was $k_{on}$ = $k_{off}$/ $K_D$ = 2.9 µM$^{-1}$ s$^{-1}$. The diffusion coefficient for Aurora B was chosen based on the known Stokes radius of the CPC complex, 10 nm (*Cormier et al., 2013*), which corresponds to D = 1 µm$^2$/s (*Luby-Phelps, 2000*). Size of the spatial interval for calculations was R = 3 µm, corresponding to the linear size of chromosome arms. Simulations for the model with no bistability in the range of physiological Aurora B concentrations were done with $k_{cis}$ = 7.3 · 10$^{-4}$ s$^{-1}$, and all other model parameters were the same as in the main model that exhibited bistability in this concentration range.

## Modeling of bistability and hysteresis experiments in cells

For bistability plots in *Figures 6B and D*, we calculated the steady-state solutions of system (6) for the indicated range of phosphatase and total Aurora B kinase concentrations (bound and soluble). The resulting total fraction of active Aurora B for each phosphatase and kinase concentrations was averaged over the entire spatial interval. The bistability region corresponds to concentrations that led to different solutions obtained using different initial conditions: Aurora B kinase active or partially active.

To model inhibition of Aurora B activity the following initial conditions were used:

$$
\begin{cases}
A^*|_{t=0} = A^{total} \\
\{A,\ B,\ B^*, [AA^*],\ [BA^*],\ [BB^*],\ [AB^*],\ [A^*PPase],\ [B^*PPase]\ \}\,|_{t=0} = 0 \\
Sites|_{t=0} = Profile1(x)
\end{cases}
\tag{8}
$$

For this simulation Aurora B autoactivation constants were calculated based on inhibitor concentration: $k_{cis}{}^{ZM} = k_{cis} \cdot \exp(-z/0.33)$ and $k^a{}_{cat}{}^{ZM} = k^a{}_{cat} \cdot \exp(-z/0.33)$, where z is ZM447439 concentration; exponential factor 0.33 was chosen in agreement with the published range 0.19 – 0.55 (*Ditchfield et al., 2003*).

In the 'inhibitor added' scenario (corresponding to 'initially high' Aurora B activity), the inhibitor was titrated from 0 μM to 2 μM with 10 nM increments. For the 'inhibitor washed out' scenario (corresponding to 'initially low' Aurora B activity), the inhibitor concentration was titrated from 2 μM to 0 μM with 10 nM decrements. For each initial condition, the steady-state fraction (at $t^{max}$ = 1,000 s) of active Aurora B kinase (bound and soluble) was calculated as:

$$
\frac{\displaystyle\int_{-R}^{R}\Big(A^*(x,\ t^{max}) + B^*(x,\ t^{max})\Big)dx}{\displaystyle\int_{-R}^{R}\Big(A(x,\ t^{max}) + B(x,\ t^{max}) + A^*(x,\ t^{max}) + B^*(x,\ t^{max})\Big)dx}
\tag{9}
$$

This approach averages the active Aurora B kinase across the entire spatial interval (*-R, R*) and reports on the total fraction of active Aurora B kinase, analogous to the normalized integrated FRET ratio in experiments with cells (*Figure 5C*).

## Modeling the propagation of Aurora B substrate phosphorylation along chromosome arms

To simulate the propagation of substrate phosphorylation after ZM447439 washout, we supplemented system (6) with the reactions of substrate phosphorylation/ dephosphorylation:

$$
\begin{cases}
\partial A^*/\partial t = F_1(x,t) - A^* \cdot S \cdot k_f + [SA^*](k_r + k_{cat}) \\
\partial B^*/\partial t = F_2(x,t) - B^* \cdot S \cdot k_f + [SB^*](k_r + k_{cat}) \\
\partial S/\partial t = ([SA^*] + [SB^*]) \cdot k_r - S \cdot (A^* + B^*) \cdot k_f + P \cdot k_P \\
\partial P/\partial t = ([SA^*] + [SB^*]) \cdot k_{cat} - P \cdot k_P \\
\partial [SA^*]/\partial t = A^* \cdot S \cdot k_f - [SA^*] \cdot k_r \\
\partial [SB^*]/\partial t = B^* \cdot S \cdot k_f - [SB^*] \cdot k_r
\end{cases}
\tag{10}
$$

where *P* and *S* are concentrations of phosphorylated and unphosphorylated substrate; other symbols are as in system (6).

The substrate was assumed to be distributed evenly over the chromatin with concentration $S_{total}$ = 1 μM, mimicking the chromatin-targeted FRET sensor. Rate constants of substrate phosphorylation/ dephosphorylation were $k_{cat} = 0.8 \cdot 10^{-3}$ s$^{-1}$ and $k_p = 0.4 \cdot 10^{-3}$ s$^{-1}$.

The following initial conditions were used:

$$
\begin{cases}
A|_{t=0} = A^{total} \\
\{A^*,\ B,\ B^*, [AA^*],\ [BA^*],\ [BB^*],\ [AB^*],\ [A^*PPase],\ [B^*PPase]\}\ |_{t=0} = 0 \\
Sites|_{t=0} = Profile2[x]
\end{cases}
\tag{11}
$$

Thus, initially Aurora B kinase was inactive, as in cells incubated with 2 μM ZM447439 (*Figure 5A and 7A*). The spatial distribution of the phosphorylated substrate *P(x,t)* was calculated along the chromosome arms axis and plotted in *Figure 7B*, and represented with color-coded images in

*Figure 7A*. To quantify the kinetics of signal propagation in *Figure 7C*, $t_0$ was set as 0 when 50% of substrate was phosphorylated at location x = 0, so $P(0,t_0) = 0.5 \cdot S_{total}$. All subsequent time points corresponded to time t when $P(x,t) = 0.5 \cdot S_{total}$.

## Modeling Aurora B activity gradient at kinetochores

The spatial distribution of the fraction of active Aurora B kinase in metaphase was calculated along the centromere-kinetochore axis using the following initial conditions:

$$\begin{cases} A^*|_{t=0} = A^{total} \\ \{A, \ B, \ B^*, [AA^*], \ [BA^*], \ [BB^*], \ [AB^*], \ [A^*PPase], \ [B^*PPase] \}|_{t=0} = 0 \\ Sites|_{t=0} = Profile1(x) \end{cases} \tag{12}$$

To achieve the steady state, the reaction-diffusion system (6) was solved numerically for time interval from 0 to $t^{max}$ = 1000 s. For each coordinate x we calculated the concentration of total Aurora B kinase (soluble and bound): $A^*(x, t^{max}) + B^*(x, t^{max}) + A(x, t^{max}) + B(x, t^{max})$ (plotted in *Figure 8—figure supplement 1C–F* in grey). Next, we calculated the fraction of active Aurora B kinase as $(A^*(x, t^{max}) + B^*(x, t^{max}))/(A^*(x, t^{max}) + B^*(x, t^{max}) + A(x, t^{max}) + B(x, t^{max}))$ and plotted it as function of x, resulting in black curves in *Figure 8—figure supplement 1C–F*. The fraction of active Aurora B kinase was also plotted in color-coded *Figure 8*, overlaid with the mesh representing local Aurora B concentration (bound and soluble). For the Aurora B activity gradient in prometaphase, a similar procedure was used, but the distance from centroid to Ndc80 was 1.75 times shorter than in metaphase. This corresponds to interkinetochore distances 1.4 µm and 0.8 µm in metaphase and prometaphase, respectively, as in HeLa cells (*Wan et al., 2009*). The distribution of Aurora B binding sites was changed accordingly, p*rofile*$^{PM}$(x) = 1.75 *profile1*(1.75 x), such that the total number of binding sites remained unchanged. Graphs in *Figure 8—figure supplement 1A,B* were calculated analogously to *Figure 6B*, but for fixed phosphatase concentration (0.1 µM). Red and blue curves in *Figure 8—figure supplement 1A,B* correspond to the initial Aurora B kinase in active or partially active forms, respectively.

## Parameter estimation and fitting tool

PEFT was used to fit experimental results for kinase autoactivation, hysteresis and bistability. This program optimized the score function value - a sum of normalized squared differences between experimental and modeled data points. The software tool was developed in Mathematica software (Wolfram Research) similarly to the systems biology software in (*Zi, 2011*) and it contained the following modules: 1) experimental data parser, 2) ODEs solver, 3) score function calculator, and 4) numerical optimizer.

### Experimental data parser

Experimental data for autoactivation experiments (*Figure 2*) were loaded into the PEFT in tables, where each table contained data from individual experiment $\{t_{ij}, D_{ij}\}$, $D_{ij}$ was the value obtained in experiment i for time point $t_{ij}$. $D_{ij}$ in each experiment was normalized to ($D_{max}$- $D_{min}$) to avoid the interference from different reactant scales when different experiments were fitted together. Each table also included metadata with the set of experimental conditions. The independent ODEs solver calculations shared same rate constants but had different initial conditions, assuring that different experiments were solved via the same system of ODEs.

### ODEs solver

To solve numerically the ODEs systems (2) and (4), the Runge-Kutta 4th order algorithm implemented in NDSolve function of Mathematica was used. Initial values for the set of model parameters $\theta = \{K_M^a, k_{cat}^a, k_{cis}\}$ for system (2) and the complete set of parameters from *Table 2* for system (4) were used. Parameter values were obtained using Numerical optimizer (see below) or set manually as needed. The initial reactant concentrations were given by the experimental data parser, as described below. ODEs solver was implemented with self-validation via the conservation laws and non-negative concentration values. The integration step was $\leq$0.02 s; we verified that calculations using 0.02 s step yielded less than $10^{-6}$ µM difference in the computed reactant concentrations relative to calculations with 0.0002 s step.

## Score function calculator

Score function was calculated as the sum of weighted sub-scores taken for all time-points of all individual experiments used in global fitting:

$$f(\theta) = \sum_{i=1}^{X} \sum_{j=1}^{T_i} \frac{\left(D_{ij} - c_i(t_{ij}, \theta)\right)^2}{\sigma_{ij}} \tag{13}$$

where $T_i$ is a number of time-points in experiment $i$, $c_i(t_{ij}, \theta)$ is the model solution corresponding to experiment $i$ obtained with specific ODEs solver instance for time point $t_{ij}$ and set of model parameters $\theta$. Weight coefficients $\sigma_{ij}$ were used to equalize the influence of experiments with different number of data points.

## Numerical optimizer

The best-fit values of model parameters were found by multidimensional optimization of the objective function (13) with the set of model parameters $\theta$. The overall goodness of model fit with experiment was assessed by the minimal score function value found by optimization (*Moles et al., 2003*; *Zi and Klipp, 2006*; *Zi, 2011*).

$$\theta^{optimal} = \mathrm{argmin}[f(\theta)] \tag{14}$$

Levenberg-Marquardt algorithm implemented in Mathematica 'FindArgMin' function was used with a default configuration and precision control. The optimization was started from random initial values of model parameters, and the good convergence was found for more than 500 launches, implying that final parameter values corresponded to the global minimum of the score function. Solutions of the system of equations (2), corresponding to the set of optimal values of the fitting parameters $\theta^{optimal}$, were displayed as solid lines in *Figures 2A and D*.

# Acknowledgements

We thank Barbara Imperiali for the Sox chemosensor, Todd Stukenberg for the Aurora B/INCENP plasmid, and Les Dutton's and Michael Ostap's labs at UPenn for use of the fluorimeters. This work was supported by National Institutes of Health grants GM083988 to MAL and ELG and GM105654 to BEB, and by grants from the Russian Fund for Basic Research (13-04-40188-H, 13-04-40190-H and 15-04-04467) and the Presidium of the Russian Academy of Sciences ('Mechanisms of the Molecular Systems Integration' and 'Molecular and Cell Biology programs') to FIA. ELG is supported in part by a Research Scholar Grant, RSG-14-018-01-CCG from the American Cancer Society. Theoretical modelling was supported by grant from Russian Science Foundation (16-14-00-224) to FIA (sections Theoretical modeling of experiments in vitro and Theoretical modeling of experiments in cells; *Figures 2*,*3*,*6–8*).

# Additional information

### Funding

| Funder | Grant reference number | Author |
| --- | --- | --- |
| National Institutes of Health | GM105654 | Ben E Black |
| Russian Foundation for Basic Research | 13-04-40188-H | Fazly I Ataullakhanov |
| Russian Academy of Sciences | Molecular and Cell Biology programs | Fazly I Ataullakhanov |
| Russian Science Foundation | 16-14-00-224 | Fazly I Ataullakhanov |
| Russian Foundation for Basic Research | 13-04-40190-H | Fazly I Ataullakhanov |
| Russian Foundation for Basic Research | 15-04-04467 | Fazly I Ataullakhanov |

| National Institutes of Health | GM083988 | Ekaterina L Grishchuk |
| American Cancer Society | RSG-14-018-01-CCG | Ekaterina L Grishchuk |

Theoretical modelling was supported by grant from Russian Science Foundation (16-14-00-224) to FIA (sections Theoretical modeling of experiments in vitro and Theoretical modeling of experiments in cells; Figures 3,6,7,8). The funders had no role in study design, data collection and interpretation, or the decision to submit the work for publication.

## Author contributions

AVZ, Theoretical modeling; DSP, Acquisition of data in vitro; MG, Acquisition of data in vitro, Theoretical modeling; AC, ERB, RS, AMM, Acquisition of data in cells; LP, BEB, Contributed unpublished essential data or reagents; FIA, Conception and design, Analysis and interpretation of data; MAL, ELG, Conception and design, Analysis and interpretation of data, Drafting or revising the article

## Author ORCIDs

Ekaterina L Grishchuk, http://orcid.org/0000-0002-9504-4270

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
