## [Decision Letter]

Thank you for submitting your work entitled "Bistability of a coupled Aurora B kinase-phosphatase system in cell division" for peer review at *eLife*. Your submission has been favorably evaluated by Vivek Malhotra (Senior editor) and three reviewers, one of whom is a member of our Board of Reviewing Editors.

The reviewers have discussed the reviews with one another and the Reviewing editor has drafted this decision to help you prepare a revised submission. In particular, the reviewers were concerned that you had not demonstrated the presence of a bistable system in vivo since any combination of a positive feedback and an inhibitor could generate a bistable condition by adjusting parameters. Therefore, demonstrating this using an unphysiological phosphatase at potentially unphysiological kinase concentrations may not be so valuable to understand the problem. Although the apparent hysteresis in response to Aurora B inhibitor is very interesting, if localization of Aurora B and relevant phosphatase are affected by Aurora B inhibitor then this phenomenon may be nothing to do with a bistability observed in vitro.

The authors "propose that bistability of an Aurora B- phosphatase system underlies formation of spatial phosphorylation patterns, generated from sites of kinase autoactivation". However, it is unclear how the authors think why this bistability helps the spatial regulation of Aurora B-dependent phosphorylation. What kind of different spatial patterns can be uniquely generated by the bistable behavior of Aurora B-dependent phosphorylation? Without critically assessing that, there seems to be a significant gap between the proposal and the evidence. These points are outlined in more detail below.

Summary:

In this study the authors have addressed the question of how Aurora B is activated and sets up a gradient of activity in the cell. The authors use a combination of mathematical modelling, biochemical reconstitution and in vivo inducible-targeting to show that Aurora B has to be concentrated at centromeres to become active, that it activates in cis, and that this behaviour is predicted from measuring Aurora B activation in vitro. They further show that in the presence of a phosphatase Aurora B activity exhibits bi-stability that depends on kinase but not phosphatase concentration. Thus the authors support the idea that concentrating Aurora B at centromeres is important to set up the Aurora B gradient, and that in the presence of a phosphatase Aurora B activity will exhibit bi-stable behaviour.

Essential revisions:

1) The authors state that the phosphatases opposing Aurora B are not known – although considerable data point toward the importance of PP1 – but λ is (obviously) not the relevant phosphatase in vivo and, given all the recent work showing how important phosphatase regulation is to mitosis, I am concerned that ignoring the identity of the opposing phosphatase or phosphatases could undermine the validity of the conclusions. Reconstituting the system using a more relevant phosphatase would be reassuring.

2) More evidence is needed to support their claim that Aurora B and phosphatase can form a bistability in a physiological context. Furthermore, even if the bistability can form at a certain context, it is far from clear if this indeed explains or supports the spatial regulation of Aurora B-dependent phosphorylation.

3) Abstract. The major conclusion, as described in the Abstract, is that Aurora B "activity at a distance depends on both sites of high kinase concentration and the bistability of a coupled kinase-phosphatase system." However, while the evidence for the sufficiency of the local kinase concentration is well supported, there is no experimental evidence that argues for or against the requirement of bistability in the spatial regulation of Aurora B kinase activity.

4) Figure 2. The mechanisms of autoactivation of the kinase were investigated using the chemosensor system, and the data are very useful in determining the mechanism of kinase autophosphorylation. However, even though it has been shown that at least two (and perhaps more) phosphorylations on the aurora-INbox construct (Sessa et al. 2005, Mol Cell, 18, 379) are necessary for full activation, the authors compare two mechanisms of auto phosphorylation (Figure 2), both of which are single phosphorylation events. Further, it is possible that the two sites are phosphorylated through different mechanisms. Indeed, structural study by Sessa et al. suggested that INCENP phosphorylation at the TSS motif, which enhances Aurora B activity, must be mediated by a *trans* mechanism. Since the *k_cat_*used here is measured by fully activated form of Aurora B-Inbox (Figure 3—figure supplement 1 and Table 1), it is not clear how fully activated A* can be achieved by the purely in cis model as in Figure 2. The conclusions drawn from the analysis of the phosphorylation curves are therefore incomplete and need further modeling. Additional experiments using phosphorylation site mutants would help resolve this pint. Furthermore, tracking down the kinetics of Aurora B phosphorylation and INbox phosphorylation (perhaps by phostag gel) would be necessary. This is important since “the dephosphorylated Aurora B kinase was two orders of magnitudes less active than the phosphorylated Aurora B", but this partially active Aurora B appears to contain substantial amount of phosphorylated INCENP (Figure 2—figure supplement 1).

Both models predict that the sensor reading is proportional to t2, and the only discerning feature between the three models is only the coefficient of t2. It would thus be more useful to fit each of the curves independently to y=k(A) t2 and then plot log (k(A)) vs A. The plot log (k(A)) vs. A should be a straight line with slope of one for the *cis* activation mechanism and two for the *trans*-activation mechanism. The plot might also indicate a fractional slope, indicating that both mechanisms are operative and further would indicate the significance of each of the mechanisms. To distinguish between these models, additional data using different Aurora B concentration are required. This approach would also avoid the complications of using the parameter obtained by fully activate kinase (*K_M_*and *V_max_* from Figure 3—figure supplement 1).

In addition, the nature of the curves may be affected by the linearity of the chemosensor. Although, it may be inferred from the Methods section, that the standard curves for the sensor are linear, the range of linearity is not mentioned, and maybe critical in fitting the initial stages of phosphorylation in Figure 2 and D.

5) The models used to fit the data shown in Figure 2 and Figure 3—figure supplement 1 are not mentioned either in the text, or in the figure legend, but from the Materials and methods section (subsection “Theoretical modeling“), the data seems to have been fitted with equation 2 or equation 1. The differential equations in both equation set 1 and equation set 2 contain processes of two vastly different time scales. Substrate binding to enzyme is usually on the order or milliseconds, and enzymatic activity is on the order of minutes in this case. Using these two time scales in the same ODE simulation will necessarily lead to very unstable results. For example, the value of *k_f_* from Table 1 is 0.38 µM-1s-1 is orders of magnitude less than usual substrate binding constants obtained from stop-flow spectrophotometry (e.g. Bowman et al. 1992, J. Biol. Chem, 267, 5346). A wrong estimate would affect the predictions of all the other systems modelled using these parameters. Since substrate concentrations are far higher than enzyme concentrations (10 nM enzyme compared to 7.5 µM sensor), the quasi-steady state approximation may be applied to substrate binding (Segel & Slemrod 1989, SIAM Review, 31, 446; Rao & Arkin 2003, J. Chem. Phys, 118, 4999) to avoid simulations on vastly different time scales and simplify the process considerably. This would mean that concentration [SA*] is constant in time. This assumption would reduce the three variables *k_f_, k_r_* and *k_cat_*to just two variables *K_M_*and *k_cat_* and reduce three differential equations to one analytical expression in eq 1.

The above stated concerns are valid to a certain extent for enzyme autophosphorylation as well. But, the quasi-steady state approximation is not valid for enzyme autophosphorylation, and other simplifications may need to be used (Schnell & Maini 2000, Bull. Math. BIol, 62, 483) while fitting data from Figure 3 and modelling for Figure 4, Figure 4—figure supplement 1 and Figure 5.

6) The data in Figure 3 (grey dots) looks to be standard sigmoid that saturates at 2 µM active kinase concentration. This is clearly to be expected since the total kinase itself added was 4 µM. The model used here doesn't account for the fact that the kinase activation reaches partial saturation (i.e. activated kinase concentration is comparable to inactive kinase). If this is rectified using a simple balance for total kinase added, it is clear that the model would predict the saturation in activated kinase seen in Figure 3.

7) Figure 4. The authors model a coupled kinase-phosphatase system using the auto phosphorylation data obtained and show that the system results in possible bistable behaviour (Figure 4). However, the models used here don't account for saturation of kinase activation (Figure 3). This would change the parameters used to fit the models, and thus changing boundaries and position of the bistable region in the bifurcation plot (Figure 4) and the quantitative predictions in Figure 4 and Figure 4—figure supplement 1A, B and C.

Even if the corrections needed to the hysteresis curves don't significantly alter the bifurcation plot in Figure 4, the authors have shown that the bistable behaviour occurs at concentrations of kinase well above physiological concentration (4 µM in comparison to 10 nM) and in a very tight range of phosphatase concentrations (0.2 µM to 0.6 µM PPase). To explain the gradients under physiological concentration using the bistability proposed by the authors, the concentration of kinase over the entire gradient would have to lie in the bistable region, i.e. greater than 4 µM. Even though it is possible that the local concentration at the centromere and the anaphase spindle midzone may possibly be high enough, it would be unlikely that the remaining kinase is enough to maintain such high concentrations over the entire gradient, especially in anaphase. The authors may argue that Aurora B becomes quickly inactivated upon diffusing out of the bistability regime, but if so, significance of bistability (or hysteric behaviour) of the Aurora B-phosphatase system in the reaction-diffusion model is not apparent. To sufficiently demonstrate significance of the bistable behaviour, the authors would need to address this issue.

8) The possible hysteresis shown in Figure 5 is interesting data, but the inactivation curve (Figure 5-red dots) shows negligible non-linearity which is concerning, since the model predictions (Figure 5-solid red line) are obviously non-linear. This casts some doubt on the bistability of the in vitro system.

9) The in vivo data shown in Figure 6 may represent the most interesting hysteretic behaviour in the paper. From the data in Figure 6, the only concentration reliably in the bistable region seems to be 0.75 µM, but the data from Figure 6 seems to expand the bistable region from 0.5 to 1 µM. This could be rectified, if instead of plotting, the averages of histograms, the average of each single cell is plotted at each inhibitor concentration as shown in Figure 2 of Ozbudak et al. (2004, Nature, 427, 737).

Do the top panels represent the sensor in Figure 6? If so, why intensities are substantially different between two images? What is the relationship between the sensor intensity and the FRET ratio?

Since the data in Figure 6 looks quite convincing, it might also be useful to increase the number of intermediate concentration points to firmly establish the bistable region in vivo.

Even if the authors address the points above, two major caveats would remain. 1) It is well established that Aurora B localization at the centromere is compromised by Aurora B kinase inhibition. If the kinase amount of the centromere changes, it is hard to argue that this is bistability, since the kinase inhibition affects not only kinase activity but also the local concentration of the kinase itself. Furthermore, PP1 at the kinetochore can be regulated by Aurora B, as shown by the Lampson lab. Thus, the process is far more complex than discussed here. 2) Correlated with the point 1, it is not obvious if the concentration of the inhibitor and the kinase activity is linear. If not, it is very difficult to prove the bistability. To examine this point, it would be critical to measure local concentration of active Aurora B (perhaps using anti-phospho Aurora B) over a wide range of Aurora B inhibitor concentrations.

These points may be addressed by targeting Aurora B-INbox to the centromere, but even so, it is still not clear if the bistable (or hysteresis) behaviour of the Aurora B-phosphatase system helps explain how kinetochore phosphorylation is controlled by microtubule attachment status.

10) Regarding the theoretical modeling of Figure 2 (subsection “Theoretical modeling”). First, the authors should state clearly that, in deriving the equation for P(t) in Figure 2, they are assuming that A*(0) = 0 (A*(t) = active form of Aurora B kinase) and there is very little net conversion of inactive A into A* during the 15 min of the experiment. With these assumptions, P(t) grows like t^2, as in the equation in Figure 2. Fine, but then the authors should check the assumption they have made in deriving the equation. For *k_cat_*_*trans* = 0.0004 /s/uM and A(0) = 0.1 uM (the yellow data in Figure 2), I get dA*/dt = 0.004 nM/s. Over the course of 16 min (1000 s), A* will increase from 0 to 4 nM, and A will decrease from 100 nM to 92 nM. Hence, the assumption that very little A is converted into A* during the course of the experiment is probably OK.

11) More troubling, though, is the following back-of-an envelope calculation of P(t) using the data in Figure 2. Using the parameter values in Table 2, I get *K_cat_*S/(*K_M_*+S) = 3/s. Using *k_cat_*_*trans**A^2 = 0.004 nM/s for the yellow curve in Figure 2 get P(t) = 22 (nM/min^2) t^2. Hence, P(5 min) = 0.55 uM and P(10 min) = 2.2 uM. These numbers are not consistent with the black curve drawn through the yellow data. My numbers are off by ~ two-fold, but I can't find the missing factor of 2.

12) There is another numerical discrepancy between Figure 5 and Figure 4—figure supplement 1. The calculated curve in Figure 4—figure supplement 1 shows that AurB decays to its steady-state activity in ~3 h, but the measured curve (Figure 5) gets to steady state much more quickly (~1 h).

[Editors' note: further revisions were requested prior to acceptance, as described below.]

Thank you for resubmitting your work entitled "Bistability of a coupled Aurora B kinase-phosphatase system in cell division" for further consideration at *eLife*. Your revised article has been favorably evaluated by Vivek Malhotra (Senior editor) and a Reviewing editor. We will be happy to accept your study if you include a short caveat to the effect that you have used a non-physiological phosphatase in your model system, and you discuss how the characteristics of the system could change in vivo given the evidence that phosphatase activity is spatially regulated in vivo.

---

## [Author Response]

*The reviewers have discussed the reviews with one another and the Reviewing editor has drafted this decision to help you prepare a revised submission. In particular, the reviewers were concerned that you had not demonstrated the presence of a bistable system in vivo since any combination of a positive feedback and an inhibitor could generate a bistable condition by adjusting parameters. Therefore, demonstrating this using an unphysiological phosphatase at potentially unphysiological kinase concentrations may not be so valuable to understand the problem.*

Our evidence for bistability of Aurora B kinase in vivo is summarized in our answer to essential point 2a; the relative unimportance of phosphatase identity is emphasized in our answer to essential point 1. We agree with the statement that “any combination of a positive feedback and an inhibitor could generate a bistable condition by adjusting parameters”, but only in a very broad, theoretical sense. As experimental scientists, we do not have access to all possible parameters, nor can we adjust them uncontrollably. Until we purified Aurora B kinase and mixed it with a phosphatase, it was not known if Aurora B activity is bistable in the range of Aurora B concentrations accessible to us. This experimental range we study in vitro (up to 8 µM) is physiological, as we now explain in our paper with Figure 6 and Figure 8, see Methods section IV Choice of model parameters. Moreover, we now provide an example of a modified kinase-phosphatase model, in which only one parameter for Aurora B intramolecular autoactivation (*k_cis_*) is modified. Although this model still includes a positive feedback and an inhibitor (opposing phosphatase), it predicts no bistable behavior in the physiological range of Aurora B concentrations (new Figure 6) and our analysis shows that this is true even for other values of enzymatic parameters for phosphatase. Consequently, this model fails to describe our experimental data (Figure 6 and Figure 7). Thus, the general statement about the behavior of certain non-linear systems does not undermine our specific findings for Aurora B kinase.

*Although the apparent hysteresis in response to Aurora B inhibitor is very interesting, if localization of Aurora B and relevant phosphatase are affected by Aurora B inhibitor then this phenomenon may be nothing to do with a bistability observed in vitro.*

See our response to essential points 2a and 9d.

*The authors "propose that bistability of an Aurora B- phosphatase system underlies formation of spatial phosphorylation patterns, generated from sites of kinase autoactivation". However, it is unclear how the authors think why this bistability helps the spatial regulation of Aurora B-dependent phosphorylation. What kind of different spatial patterns can be uniquely generated by the bistable behavior of Aurora B-dependent phosphorylation? Without critically assessing that, there seems to be a significant gap between the proposal and the evidence. These points are outlined in more detail below.*

See our response to essential point 2b, 7b and 9f.

*Summary: In this study the authors have addressed the question of how Aurora B is activated and sets up a gradient of activity in the cell. The authors use a combination of mathematical modelling, biochemical reconstitution and in vivo inducible-targeting to show that Aurora B has to be concentrated at centromeres to become active, that it activates* in cis*, and that this behaviour is predicted from measuring Aurora B activation in vitro. They further show that in the presence of a phosphatase Aurora B activity exhibits bi-stability that depends on kinase but not phosphatase concentration. Thus the authors support the idea that concentrating Aurora B at centromeres is important to set up the Aurora B gradient, and that in the presence of a phosphatase Aurora B activity will exhibit bi-stable behaviour. Essential revisions: 1) The authors state that the phosphatases opposing Aurora B are not known* – *although considerable data point toward the importance of PP1* –

*but λ is (obviously) not the relevant phosphatase in vivo and, given all the recent work showing how important phosphatase regulation is to mitosis, I am concerned that ignoring the identity of the opposing phosphatase or phosphatases could undermine the validity of the conclusions. Reconstituting the system using a more relevant phosphatase would be reassuring.*

Considerable evidence points to PP1 acting on the same cellular substrates as Aurora B kinase, but we are not aware of strong evidence that this phosphatase regulates the Aurora B/INCENP complex in cells. Both PP1 and PP2A have been associated with Aurora B (e.g., Sugiyama et al. 2002, Oncogene), but their relative importance and the relevant regulatory subunits are not known. The regulatory subunits are particularly important because they provide substrate specificity for the catalytic subunits, so using the catalytic subunit alone would not make the reconstitution more physiological. Additionally, the phenomenon of bi-stability in our coupled kinase-phosphatase system does not necessarily depend on the identity or specific characteristics of the phosphatase. Indeed, our theoretical analysis (“Analysis of bistability dependence on catalytic constants for phosphatase” subsection of part III of the Materials and methods: Theoretical modeling of experiments in vitro) demonstrates that the kinetics of Aurora B kinase autoactivation is the driving force for this bi-stable system, while phosphatase characteristics are not as important. As one reviewer pointed out, the phosphatase in our reconstitution simply provides “friction for the kinase to work against”, so we believe that the current lack of knowledge about the identity of physiological phosphatase is not detrimental. Finally, we are not aware of any published in-depth analysis, theoretical or via reconstitution, of bistability of Aurora B kinase with any phosphatase, so our work discovering and characterizing this phenomenon represents a conceptually novel advance. As we had to prioritize our efforts for resubmission within the given time constraint, we decided to focus on other issues, where our efforts could make a larger impact.

2) More evidence is needed to support their claim that Aurora B and phosphatase can form a bistability in a physiological context.

a) We believe that the evidence we already presented for bistability of Aurora B-dependent phosphorylation in vivo is quite strong, because it clearly shows two distinct states for phosphorylation present simultaneously at identical conditions in a cell population (previous Figure 6, now Figure 5). We also show that a reconstituted system of only two components, Aurora and phosphatase, exhibits the same behavior (bistability and hysteresis) in vitro in a similar range of Aurora B concentrations. Our logic is simply that bistability in cells could reflect an intrinsic property of Aurora B kinase opposed by some cellular phosphatase(s). This hypothesis is not only the most straightforward, but we put it forward in the absence of any other model that could explain bistability and hysteresis observed in cells. Indeed, these non-linear phenotypes strongly imply that the underlying molecular reactions are also bistable. The reviewers have suggested that changes in localization of kinase or phosphatase at different ZM447439 concentration could be responsible. Our experiments, however, have not found any changes in localization of these components (new Figure 5—figure supplement 1, see also critical point 9d). An additional hypothesis suggested by the reviewers was that bistability in cells could be due to a specific dependency between ZM447439 concentration and Aurora kinase activity, but we argue that this also cannot explain our cellular results (see critical point 9e). Finally, our newly added quantification of the propagation of Aurora B activity waves following ZM447439 washout shows that it is characterized by a constant speed, a hallmark feature of bistable systems, which we have now documented in a physiological context (i.e., in living cells) (new Figure 7).

*Furthermore, even if the bistability can form at a certain context, it is far from clear if this indeed explains or supports the spatial regulation of Aurora B-dependent phosphorylation.*

b) We had proposed this idea based on general knowledge of the spatial behavior of bistable systems (Liehr, Dissipative solutions in reaction-diffusion systems, 2013, Springer) and on our preliminary modeling. We have now built a mechanistic molecular model for spatial regulation of Aurora B activity in the cell, incorporating the enzymatic constants we measured for Aurora B autoactivation in vitro (see new sections in main text). New experiments were conducted to collect some missing parameter values for this model (the distribution of Aurora B along chromosome arms in new Figure 6—figure supplement 1, panels B, C). First, we demonstrate that this model is quantitatively consistent with bistibility and hysteresis observed in experiments in cells with varying ZM447439 concentrations (new Figure 6). Second, we modeled two spatial phenomena: propagation of Aurora B phosphorylation along chromosome arms and distribution of active Aurora B kinase at the kinetochore in prometaphase and metaphase. Next, we measured the rate of phosphorylation propagation in live cells and found that it is constant (indicating that the underlying molecular system is bistable) and quantitatively consistent with model prediction (Figure 7). Moreover, a variation on our model, which included Aurora B kinase autoactivation (positive feedback) and opposing phosphatase but not bistability in the physiological concentration range, failed to describe this experiment. Finally, our modeling of the distribution of Aurora B activity at the centromere and kinetochore clearly demonstrates that bistability of an Aurora B kinase-phosphatase coupled system is essential to establish a steep gradient of Aurora B activity at the sites where its microtubule-binding substrates are located (new Figure 8 and Figure 8—figure supplement 1). The lack of a quantitative explanation for how a gradient with such properties is established has been previously identified in this field as one of the major unresolved questions (Krennand Musacchio, 2015, Front. Oncol). Our findings therefore provide a conceptually novel mechanism, which future work will examine and test.

3) Abstract. The major conclusion, as described in the Abstract, is that Aurora B "activity at a distance depends on both sites of high kinase concentration and the bistability of a coupled kinase-phosphatase system." However, while the evidence for the sufficiency of the local kinase concentration is well supported, there is no experimental evidence that argues for or against the requirement of bistability in the spatial regulation of Aurora B kinase activity.

With our new results, the evidence supporting this statement is now more convincing.

*4) Figure 2. The mechanisms of autoactivation of the kinase were investigated using the chemosensor system, and the data are very useful in determining the mechanism of kinase autophosphorylation. However, even though it has been shown that at least two (and perhaps more) phosphorylations on the aurora-INbox construct (Sessa et al.2005, Mol Cell, 18, 379) are necessary for full activation, the authors compare two mechanisms of auto phosphorylation (Figure 2), both of which are single phosphorylation events. Further, it is possible that the two sites are phosphorylated through different mechanisms. Indeed, structural study by Sessa et al. suggested that INCENP phosphorylation at the TSS motif, which enhances Aurora B activity, must be mediated by a trans mechanism. Since the k_cat_ used here is measured by fully activated form of Aurora B-Inbox (Figure 3—figure supplement 1 and Table 1), it is not clear how fully activated A* can be achieved by the purely* in cis *model as in Figure 2. The conclusions drawn from the analysis of the phosphorylation curves are therefore incomplete and need further modeling. Additional experiments using phosphorylation site mutants would help resolve this pint. Furthermore, tracking down the kinetics of Aurora B phosphorylation and INbox phosphorylation (perhaps by phostag gel) would be necessary.*

a) Experiments with different phosphosite mutants would help to clarify the exact role for different phosphorylation sites, but since this is not the main point of our paper we followed the reviewers’ suggestion to improve our analysis computationally. In the previous version of the manuscript we used data in old Figure 2 to make two important points.

First, we proposed a previously overlooked *cis*-component in Aurora B autoactivation. This finding is important because, as we show in the new manuscript, the weak *cis* component provides bistability in the physiological range of Aurora B concentrations. To address the criticism that this conclusion needs additional analysis, we have now confirmed the *cis*-autoactivation mechanism using the alternative approach suggested by the reviewers (see critical point 4c), instead of using fitting with the equation in old Figure 2.

Second, we previously used this equation to estimate enzymatic parameters for *cis* Aurora B autoactivation. In the current manuscript we reach this goal by a different approach: using global fitting to data in new Figure 2 panels A and D (see section Determination of parameters of Aurora B autoactivation in the Materials and methods). This new approach resulted in slightly different estimated constants, but all main conclusions of our study remain valid. Please also note that due to severe time constrain for resubmission, we were unable to synthesize sufficient quantities of the custom sox-sensor which we used in the previous manuscript version. All experiments reported in new Figure 2 have been repeated with a commercial Omnia sensor, which is phosphorylated with different constants relative to our own chemosensor. All these technical details are described in Materials and methods sections II and III and all constants are listed in renewed Table 1.

Finally, we agree that our modeling approach was simplified with the assumption of a single phosphorylation site. To satisfy this criticism we have built and analyzed a model that incorporates two sites for Aurora B activation (new Methods section Model of Aurora B autoactivation via two phosphorylation sites, Table 3 and Figure 2—figure supplement 3). As expected, we can fit all experimental data with the two sites model. Importantly, these new results justify our use of a simplified model with only one phosphorylation site, as the one-site model can be interpreted as a two-site model with a rapidly converting Aurora B form with one phosphorylated site.

*This is important since “the dephosphorylated Aurora B kinase was two orders of magnitudes less active than the phosphorylated Aurora B", but this partially active Aurora B appears to contain substantial amount of phosphorylated INCENP (Figure 2—figure supplement 1).*

b) Old Figure 2—figure supplement 1 does not show what we have called “partially active Aurora B”; we previously made a mistake citing this figure in this context. Instead, it is a control for old Figure 2 and Figure 3, where Aurora B autoactivation was examined in the presence < 100 nM phosphatase and 10 mM phosphatase inhibitor. The figure was included to show that this inhibitor concentration is sufficient to fully block the activity of phosphatase towards INbox. To alleviate the reviewers’ concern, we have added new Figure 2—figure supplement 1, which shows that Inbox associated with the partially active Aurora B is completely unphosphorylated.

*Both models predict that the sensor reading is proportional to t2, and the only discerning feature between the three models is only the coefficient of t2. It would thus be more useful to fit each of the curves independently to y=k(A) t2 and then plot log (k(A)) vs A. The plot log (k(A)) vs. A should be a straight line with slope of one for the cis activation mechanism and two for the trans-activation mechanism. The plot might also indicate a fractional slope, indicating that both mechanisms are operative and further would indicate the significance of each of the mechanisms. To distinguish between these models, additional data using different Aurora B concentration are required. This approach would also avoid the complications of using the parameter obtained by fully activate kinase (K_M_ and V_max_ from Figure 3—figure supplement 1).*

c) The reviewers likely mean that log(k(A)) should be plotted vs. logA (not vs. A) to obtain a straight line with the slope of 1 for *cis*-activation and 2 for the *trans*-activation mechanism. First, we obtained new experimental curves for a wider range of Aurora B concentrations (0.16 – 1.5 µM), as suggested by the reviewers. Second, these curves were analyzed using the suggested approach, and our conclusion has not changed. Indeed, new Figure 2 shows a line with the slope value 1.23 ± 0.02, which demonstrates that *cis*-activation is dominating over *trans*-activation during initial activation at low kinase concentration. Additionally, we now plot calculated fraction of Aurora B kinase phosphorylated in cis vs. in trans, and the resulting product concentrations (see new Figure 2—figure supplement 2 panels H and I), so our point is more clear.

*In addition, the nature of the curves may be affected by the linearity of the chemosensor. Although, it may be inferred from the Methods section, that the standard curves for the sensor are linear, the range of linearity is not mentioned, and maybe critical in fitting the initial stages of phosphorylation in Figure 2 and D.*

d) In general, fluorescence is linear to concentration of dye in a very large concentration range (up to 100 mM). The calibration curves for two chemosensors are now provided for the concentration range used in our study (new Figure 2—figure supplement 2).

*5) The models used to fit the data shown in Figure 2 and Figure 3—figure supplement 1 are not mentioned either in the text, or in the figure legend, but from the Materials and methods section (subsection “Theoretical modeling“), the data seems to have been fitted with equation 2 or equation 1.*

a) We are sorry that this information was not visible enough. The fitting strategy in the new manuscript version has been simplified; the number of figures for in vitro analysis was reduced. All fitting details are described in a significantly improved manner in the Materials and methods.

*The differential equations in both equation set 1 and equation set 2 contain processes of two vastly different time scales. Substrate binding to enzyme is usually on the order or milliseconds, and enzymatic activity is on the order of minutes in this case. Using these two time scales in the same ODE simulation will necessarily lead to very unstable results. For example, the value of k_f_ from Table 1 is 0.38 µM-1s-1 is orders of magnitude less than usual substrate binding constants obtained from stop-flow spectrophotometry (e.g. Bowman et al. 1992, J. Biol. Chem, 267, 5346). A wrong estimate would affect the predictions of all the other systems modelled using these parameters. Since substrate concentrations are far higher than enzyme concentrations (10 nM enzyme compared to 7.5 µM sensor), the quasi-steady state approximation may be applied to substrate binding (Segel & Slemrod 1989, SIAM Review, 31, 446; Rao & Arkin 2003, J. Chem. Phys, 118, 4999) to avoid simulations on vastly different time scales and simplify the process considerably. This would mean that concentration [SA*] is constant in time. This assumption would reduce the three variables k_f_, k_r_ and k_cat_ to just two variables K_M_ and k_cat_ and reduce three differential equations to one analytical expression in eq 1.*

b) Indeed, equation sets 1 and 2 describe processes of different time scales; however, we believe that our calculation algorithm deals with this issue properly. To demonstrate that the obtained numerical solutions are correct, we have recalculated these equation sets using a reduced time step. Specifically, we verified that calculations using 0.02 s step yielded less than 10-6 µM difference relative to calculations with 2 × 10-4 s step). Furthermore, we applied a quasi-steady state approximation, as suggested by the reviewers, to describe the data in new Figure 2 and obtained the same results as with our calculation scheme.

*The above stated concerns are valid to a certain extent for enzyme autophosphorylation as well. But, the quasi-steady state approximation is not valid for enzyme autophosphorylation, and other simplifications may need to be used (Schnell & Maini 2000, Bull. Math. BIol, 62, 483) while fitting data from Figure 3 and modelling for Figure 4, Figure 4—figure supplement 1 and Figure 5.*

c) We agree that the quasi-steady state approximation cannot be used for analysis of different schemes of kinase autoactivation with no substrate because different schemes require different simplifications. To avoid this complication, we used a sophisticated numerical approach (see Materials and methods section V), which solves all equation sets in our paper in a unified and consistent manner.

*6) The data in Figure 3 (grey dots) looks to be standard sigmoid that saturates at 2 µM active kinase concentration. This is clearly to be expected since the total kinase itself added was 4 µM. The model used here doesn't account for the fact that the kinase activation reaches partial saturation (i.e. activated kinase concentration is comparable to inactive kinase). If this is rectified using a simple balance for total kinase added, it is clear that the model would predict the saturation in activated kinase seen in Figure 3.*

We think the reviewers will agree that the experimental data in the previous Figure 3 was noisy, and the measurements stopped before the plateau could be observed convincingly. This experiment is technically difficult because 1) to conserve reagents the starting reaction is only 8-16 µl in volume and 1-2 µl samples are taken at different times to access kinase activity, and 2) the reaction and sampling take place in a dialysis tube to avoid depletion of ATP and possible ADP inhibition. We have improved our technique and repeated this experiment more accurately with larger statistics and longer incubation times. The improved plot (new Figure 2) shows saturation at the maximum concentration.

*7) Figure 4. The authors model a coupled kinase-phosphatase system using the auto phosphorylation data obtained and show that the system results in possible bistable behaviour (Figure 4). However, the models used here don't account for saturation of kinase activation (Figure 3). This would change the parameters used to fit the models, and thus changing boundaries and position of the bistable region in the bifurcation plot (Figure 4) and the quantitative predictions in Figure 4 and Figure 4—figure supplement 1A, B and C.*

a) This issue has been addressed in essential point 6 (above). The experiment has been repeated to show a lack of partial saturation (new Figure 2).

*Even if the corrections needed to the hysteresis curves don't significantly alter the bifurcation plot in Figure 4, the authors have shown that the bistable behaviour occurs at concentrations of kinase well above physiological concentration (4 µM in comparison to 10 nM) and in a very tight range of phosphatase concentrations (0.2 µM to 0.6 µM PPase). To explain the gradients under physiological concentration using the bistability proposed by the authors, the concentration of kinase over the entire gradient would have to lie in the bistable region, i.e. greater than 4 µM. Even though it is possible that the local concentration at the centromere and the anaphase spindle midzone may possibly be high enough, it would be unlikely that the remaining kinase is enough to maintain such high concentrations over the entire gradient, especially in anaphase. The authors may argue that Aurora B becomes quickly inactivated upon diffusing out of the bistability regime, but if so, significance of bistability (or hysteric behaviour) of the Aurora B-phosphatase system in the reaction-diffusion model is not apparent. To sufficiently demonstrate significance of the bistable behaviour, the authors would need to address this issue.*

b) 10 nM is the estimated concentration of soluble Aurora B (Mahen et al., 2014, MBoC), while the estimated concentration of centromere-bound kinase is significantly higher (new Figure 6—figure supplement 1). This range is similar to what we use in our in vitro experiments. We explain how the concentration profile in this figure was generated in Materials and methods section IV. Importantly, Aurora B kinase is tethered to chromatin via an elongated INCENP (Samejima et al., 2015, JBC), so one kinase molecule can reach other molecules located 30 nm away or farther. Thus, the bound kinase is likely to be able to activate in trans in a concentration dependent manner. This becomes a very complex system, because chromatin-associated kinase is also in a dynamic equilibrium with a soluble pool. According to our calculations, chromatin bound kinase becomes completely active (on state) throughout the centromeric chromatin in prometaphase. In metaphase, when the centromere is stretched and the local kinase density is decreased, the model predicts an area with bistability near the outer kinetochore region, where a steep activity gradient is predicted (new Figure 8—figure supplement 1). We demonstrate the significance of the bistable behavior by considering a model with modified Aurora B kinase, in which bistability is observed at concentrations that exceed those that are found at the centromere (new Figure 6). This model also predicts a gradient of Aurora B kinase activity, which simply reflects the distribution of its binding sites (new Figure 8—figure supplement 1). Importantly, the gradient in the model with no bistability lacks the steep region coinciding with the outer kinetochore. Moreover, this gradient does not change significantly in response to kinetochore stretching in metaphase, while the bistable model shows the expected loss of phosphorylation at the outer kinetochore in metaphase. We believe that these new data illustrate clearly the essence of a new concept that we propose for spatial Aurora B kinase regulation, in which bistability plays an essential role.

*8) The possible hysteresis shown in Figure 5 is interesting data, but the inactivation curve (Figure 5-red dots) shows negligible non-linearity which is concerning, since the model predictions (Figure 5-solid red line) are obviously non-linear. This casts some doubt on the bistability of the in vitro system.*

We added new data points to this curve (new Figure 4), and the non-linear behavior for the experimental data is now more apparent. We note that these experiments are laborious, with most data points representing at least 2 repetitions of the kinetic curves, as in new Figure 4; importantly, they consume large quantities of kinase and other reagents. These constraints preclude us from incorporating additional measurements on this already well-defined curve.

*9) The in vivo data shown in Figure 6 may represent the most interesting hysteretic behaviour in the paper. From the data in Figure 6, the only concentration reliably in the bistable region seems to be 0.75 µM, but the data from Figure 6 seems to expand the bistable region from 0.5 to 1 µM. This could be rectified, if instead of plotting, the averages of histograms, the average of each single cell is plotted at each inhibitor concentration as shown in Figure 2 of Ozbudak et al. (2004, Nature, 427, 737).*

a) Thank you for this suggestion. The data have been replotted and are now shown in new Figure 5.

*Do the top panels represent the sensor in Figure 6? If so, why intensities are substantially different between two images? What is the relationship between the sensor intensity and the FRET ratio?*

b) The top panels are CFP images showing the localization of the sensor, with higher intensity indicating a higher expression level of the sensor. This issue has come up in our previous work and publications with this sensor, and we have previously analyzed cells with a range of expression levels to find that the FRET ratio is independent of the expression level (not shown). To make the result clearer, we selected cells with similar expression levels for this figure (new Figure 5).

*Since the data in Figure 6 looks quite convincing, it might also be useful to increase the number of intermediate concentration points to firmly establish the bistable region in vivo.*

c) Our main focus during this resubmission was on examining the importance of bistability for spatial regulation of kinase signaling, but we also added an additional data point to this plot, which agrees well with the predictions of our new model (new Figure 5 and Figure 6).

*Even if the authors address the points above, two major caveats would remain. 1) It is well established that Aurora B localization at the centromere is compromised by Aurora B kinase inhibition. If the kinase amount of the centromere changes, it is hard to argue that this is bistability, since the kinase inhibition affects not only kinase activity but also the local concentration of the kinase itself. Furthermore, PP1 at the kinetochore can be regulated by Aurora B, as shown by the Lampson lab. Thus, the process is far more complex than discussed here.*

d) We note that one of the original papers reporting chemical inhibition of Aurora B showed that the kinase localizes normally in the presence of the inhibitor (Ditchfield et al. 2003, JCB, Figure 6). Consistent with this result, we now show that kinase localization does not change in our experiment (new Figure 5—figure supplement 1). Furthermore, PP1 does not localize to kinetochores in the presence of nocodazole (Liu et al., 2010 JCB), which is how our experiments showing hysteresis in vivo were performed, and we do not detect any change in PP1 localization in the presence of ZM447439 (new Figure 5—figure supplement 1).

*2) Correlated with the point 1, it is not obvious if the concentration of the inhibitor and the kinase activity is linear. If not, it is very difficult to prove the bistability. To examine this point, it would be critical to measure local concentration of active Aurora B (perhaps using anti-phospho Aurora B) over a wide range of Aurora B inhibitor concentrations.*

e) This is a good point, and the inhibitor-kinase relationship has already been published (Ditchfield et al. 2003, JCB). In this work the authors report similar IC50 for ZM447439 inhibition of Aurora A and Aurora B, and Aurora A kinase activity versus ZM447439 concentration was found to represent an exponent. We have used this dependency in our newly developed theoretical model (described in the Materials and methods) and the model predicts bistability. Importantly, a variant of this model that used this same inhibitor-kinase relationship, but a different *k_cis_* for Aurora B, predicted no bistability region for physiological Aurora B kinase concentration (new Figure 6), implying that bistability is not caused by this inhibitor-kinase relationship. Only the model with bistability provided good agreement with in vivo experimental results (new Figure 6, Figure 7).

*These points may be addressed by targeting Aurora B-INbox to the centromere, but even so, it is still not clear if the bistable (or hysteresis) behaviour of the Aurora B-phosphatase system helps explain how kinetochore phosphorylation is controlled by microtubule attachment status.*

f) We now provide new evidence for the importance of bistability in spatial regulation of Aurora B activity (new Figure 8, Figure 8—figure supplement 1 and corresponding text). Demonstrating unequivocally that this mechanism operates in cells to control microtubule attachment status is a long-term goal that is beyond the scope of the current paper. Since the bar for explaining the spatial regulation of Aurora B in cells, one of the most enigmatic and ill-understood aspects of mitosis, is very high, our resubmitted work will not be able to address such criticism in its entirety. We believe, however, that our work does represent a major advance towards addressing this important problem. As an analogy, spatial gradients of Aurora B activity in anaphase were first reported by Lampson and colleagues (Fuller et al. 2008, Nature) years before the functional significance of such gradients were revealed (Uehara et al. 2013, JCB; Ferreira et al. 2013, JCB; Afonso et al. 2014, Science). We hope you will consider our revised manuscript without a comprehensive explanation of the spatial regulation of Aurora B in vivo. Importantly, our theoretical model based on the bistability of an Aurora B kinase – phosphatase system does provide a unified explanation of our combined data both in vitro and in vivo. We have not been able to formulate an alternative hypothesis that would link together all our data and previously published results about Aurora B localization and regulation. Therefore, we believe that our current findings represent a major advance toward the long-term goal of understanding how microtubule attachments are controlled.

*10) Regarding the theoretical modeling of Figure 2 (subsection “Theoretical modeling”). First, the authors should state clearly that, in deriving the equation for P(t) in Figure 2, they are assuming that A*(0) = 0 (A*(t) = active form of Aurora B kinase) and there is very little net conversion of inactive A into A* during the 15 min of the experiment. With these assumptions, P(t) grows like t^2, as in the equation in Figure 2. Fine, but then the authors should check the assumption they have made in deriving the equation. For kcat_trans = 0.0004 /s/uM and A(0) = 0.1 uM (the yellow data in Figure 2), I get dA*/dt = 0.004 nM/s. Over the course of 16 min (1000 s), A* will increase from 0 to 4 nM, and A will decrease from 100 nM to 92 nM. Hence, the assumption that very little A is converted into A* during the course of the experiment is probably OK.*

We agree with the reviewers that to derive equations for P(t) in the previous Figure 2, we assumed that there was very little net conversion of partially active A into A* during the first 15 min. We have now changed our strategy to analyze these data (current Figure 2—figure supplement 2), so this assumption is no longer essential. We also note that we have directly calculated that in these experiments, full substrate phosphorylation takes place while active kinase constitutes less than 2% from total, so there is indeed very little conversion (Figure 2—figure supplement 2 panels H nd I).

*11) More troubling, though, is the following back-of-an envelope calculation of P(t) using the data in Figure 2. Using the parameter values in Table 1 get K_cat_S/(K_M_+S) = 3/s. Using k_cat__trans*A^2 = 0.004 nM/s for the yellow curve in Figure 2 get P(t) = 22 (nM/min^2) t^2. Hence, P(5 min) = 0.55 uM and P(10 min) = 2.2 uM. These numbers are not consistent with the black curve drawn through the yellow data. My numbers are off by ~ two-fold, but I can't find the missing factor of 2.*

This calculation did not account for a factor 2 in the equation for P(t) (previous Figure 2 and new Figure 2—figure supplement 2). Taking this factor into account results in a good match with experimental data:

*K_cat_*S / (*K_M_*+S) = 2.3s^-1 = 138 min^-1

*k_cat__cis**A = 0.004 nM s^-2 = 0.24 nM min^-1

P(t) = 33 / 2 (nM min^-2) t^2 = 16 (nM min^-2) t^2

P(5 min) = 0.4 µM

P(10 min) = 1.6 µM

Please note that in response to other critical comments, these experimental data have changed, and we no longer use the plots and equations as in the previous Figure 2 to derive enzymatic constants.

*12) There is another numerical discrepancy between Figure 5 and Figure 4—figure supplement 1. The calculated curve in Figure 4—figure supplement 1 shows that AurB decays to its steady-state activity in ~3 h, but the measured curve (Figure 5) gets to steady state much more quickly (~1 h).*

There is no discrepancy between the previous Figure 5 and Figure 4—figure supplement 1. Rate constants for Aurora B activity decay is 0.02 min_-1_ for old Figure 5 middle and 0.022 min-1 for old Figure 4—figure supplement 1A. This small difference is due to different phosphatase concentrations (0.35 µM vs. 0.4 µM).

[Editors' note: further revisions were requested prior to acceptance, as described below.]

We will be happy to accept your study if you include a short caveat to the effect that you have used a non-physiological phosphatase in your model system […]

In response to your suggestion to “include a short caveat to the effect that you have used a non-physiological phosphatase in your model system”, we have added text to:

1) State explicitly that λ phage phosphatase was used in our simplified reconstitution system in vitro (subsection “Aurora B kinase-phosphatase bistability and hysteresis observed in vitro are in a quantitative agreement with theoretical prediction“).

2) Emphasize that this phosphatase is not physiological in Discussion. The relevant section is quoted below:

“Importantly, our theoretical analyses predicted that bistability depends strongly on the Aurora B kinase autoactivation mechanism, while kinetic constants for the dephosphorylation reaction have much less impact (Figure 3—figure supplement 1). Thus, although our simplified reconstitution in vitro used the non-physiological λ phosphatase, the model suggested that these nonlinear regimes could exist in a physiological cell context, where Aurora B kinase is coupled with its native phosphatase partner(s). Indeed, we were able to recreate bistability and hysteresis for Aurora B substrate phosphorylation in live mitotic cells…”

[…] And you discuss how the characteristics of the system could change in vivo given the evidence that phosphatase activity is spatially regulated in vivo.

In response to your suggestion to “discuss how the characteristics of the system could change in vivo given the evidence that phosphatase activity is spatially regulated in vivo” we expanded the final paragraph of the Discussion section to include:

“In addition to the bistable system described here, other mechanisms may also regulate spatial patterns of Aurora B activity, such as changes in Aurora B enrichment at centromeres as chromosomes align (Salimian et al., 2011) and localized phosphatase activity at different cellular locations, such as kinetochores or centromeres or on chromatin (Trinkle-Mulcahy et al., 2003; Kitajima et al., 2006; Riedel et al., 2006; Tang et al., 2006; Trinkle-Mulcahy et al., 2006; Liu et al., 2010; Foley et al., 2011). Localization of both PP1 and PP2A at kinetochores depends on microtubule attachment and tension, and changes in these local phosphatase activities may modulate the location of the bistable region of Aurora B activity or exert direct effect on the substrates that are located in immediate vicinity.”